# Theoretical formulation of chemical equilibrium under vibrational strong coupling

Kaihong Sun[1] & Raphael F. Ribeiro [1] ✉

Experiments have suggested that strong interactions between molecular ensembles and infrared microcavities can be employed to control chemical equilibria. Nevertheless, the primary mechanism and key features of the effect remain largely unexplored. In this work, we develop a theory of chemical equilibrium in optical microcavities, which allows us to relate the equilibrium composition of a mixture in different electromagnetic environments. Our theory shows that in planar microcavities under strong coupling with polyatomic molecules, hybrid modes formed between all dipole-active vibrations and cavity resonances contribute to polariton-assisted chemical equilibrium shifts. To illustrate key aspects of our formalism, we explore a model $S_N2$ reaction within a single-mode infrared resonator. Our findings reveal that chemical equilibria can be shifted towards either direction of a chemical reaction, depending on the oscillator strength and frequencies of reactant and product normal modes. Polariton-induced zero-point energy changes provide the dominant contributions, though the effects in idealized single-mode cavities tend to diminish quickly as the temperature and number of molecules increase. Our approach is valid in generic electromagnetic environments and paves the way for understanding and controlling chemical equilibria with microcavities.

Light–matter interactions are often irrelevant in equilibrium thermodynamics[1,2]. However, recent experiments have suggested otherwise, that the chemical equilibrium of aromatic-halogen charge-transfer complexes may be significantly changed via strong light–matter coupling[3].

The signature of strong light–matter interactions is the formation of hybrid states referred to as polaritons, consisting of a superposition of electromagnetic (EM) and matter excitations[4–6]. Devices that confine the EM field to the scale of relevant wavelengths [e.g., for infrared (IR) strong coupling, planar cavities are generally constructed with moderate quality mirrors separated by a distance of O ($\mu$m)][7–11] are generally conducive to polariton formation in the presence of a resonant material (Fig. 1). A simple paradigmatic model of this

phenomenon includes an isolated cavity mode under strong interaction with the collective polarization of a molecular system containing $N$ identical molecules. This system has two hybrid light–matter modes denoted lower and upper polaritons (LP and UP, respectively), and $N-1$ molecular reservoir modes with zero photonic content.

Recent experimental reports have provided evidence that chemical reactions can be substantially affected by strong interactions between IR microcavities and near-resonant molecular vibrational normal modes (vibrational strong coupling)[12–19]. Charge conductivity[20–22] and energy transport modulation[23–27] have also been reported.

While theoretical investigations have proposed hypothetical mechanisms for microcavity effects on reaction rates via nonequilibrium effects[28–33], less attention has been paid to polariton

[1]Department of Chemistry and Cherry Emerson Center for Scientific Computation, Emory University, Atlanta, GA 30322, USA.
✉ e-mail: raphael.ribeiro@emory.edu

**Fig. 1 | Schematic representation of a reactive mixture in an infrared micro-cavity supporting confined electromagnetic field modes and strong light–matter interactions.** This setup implies the formation of hybrid polariton normal modes with distinct spectra relative to the molecular system in free space and the empty microcavity.

effects on thermodynamic quantities of molecular systems. Scholes et al.[34] showed the free energy of dark modes is lower than the polaritonic, and Li et al.[35] employed classical statistical mechanics to argue that collective strong light–matter coupling is unlikely to affect molecular potentials of mean force. However, recent quantum approaches have shown that polariton effects on thermodynamic quantities could be significant, especially under ultrastrong coupling conditions[36,37].

In this work, we present a quantum theoretical investigation of chemical equilibrium under vibrational strong coupling (VSC). We provide a theory of nonperturbative light–matter interaction effects on chemical equilibrium and obtain a mathematical relationship between the composition of equilibrium reactive mixtures inside and outside microcavities. As an example, we apply our theory to a multi-component reactive mixture in a single-mode IR cavity resonant with a bright normal mode of reactants or products. We examine the temperature, normal mode frequency, oscillator strength, and size dependence of the polariton effect on the reactive mixture composition at equilibrium. At the end, we summarize our main results and explain how the provided formalism informs future work.

## Results

In this section, we present a general formalism for the investigation of nonperturbative light–matter interaction effects on the composition of reactive mixtures. Let A, B, C, and E denote reactive chemical species in equilibrium (in the gas phase for simplicity) according to

$$\nu_A\,A + \nu_B\,B \rightleftharpoons \nu_C\,C + \nu_E\,E. \tag{1}$$

The total molecular quantum electrodynamic Hamiltonian[38] for this system in the Coulomb gauge is given by

$$H = H_M + H_L + H_{LM}, \tag{2}$$

where $H_L$ is the transverse EM field Hamiltonian that generates the free field dynamics. Without loss of generality, we assume the bare field dynamics conserves momentum along the $x$, $y$ and $z$ directions, so $H_L$ is given by

$$H_L = \sum_{\mathbf{k}\lambda}^{k<k_M} \hbar\omega_{\mathbf{k}}\left(a_{\mathbf{k}\lambda}^{\dagger}a_{\mathbf{k}\lambda} + \tfrac{1}{2}\right) \tag{3}$$

where $\mathbf{k} = (k_x, k_y, k_z)$ is the wave vector with components $k_x$, $k_y$, and $k_z$ fulfilling boundary conditions associated with the electromagnetic

environment[39], $k = |\mathbf{k}|$, $\lambda = 1, 2$ denotes the field polarization, and $k_M$ is a high-energy cutoff for photon modes. Specifically, in the treatment detailed below, only photon modes with $k < k_M$ are assumed to form polaritons. In any particular application $k_M$ would depend on the molecular system considered and the strength of the collective light–matter interaction[40]. Note other photonic structures could also be treated with Eq. (3) by employing modes defined in terms of suitable quantum numbers according to symmetry and boundary conditions satisfied by the EM field.

The pure matter part of the Hamiltonian is denoted by $H_M$ and given by

$$H_M = h_M + V_M, \tag{4}$$

where $V_M$ corresponds to the (longitudinal) intermolecular electro-static interactions (between electronic and nuclear charges of different molecules), and $h_M$ is the Hamiltonian for a noninteracting mixture of A, B, C, and E molecules

$$h_M = h_A + h_B + h_C + h_E. \tag{5}$$

The noninteracting subsystem Hamiltonian $h_F$ corresponds to the nonrelativistic electrostatic Hamiltonian describing a pure ensemble of $N_F$ noninteracting molecules of type $F \in \{A, B, C, E\}$, i.e., $h_F = \sum_{i=1}^{N_F} h_{iF}$, where

$$h_{iF} = \sum_{\alpha} \frac{\mathbf{p}_{i\alpha}^2}{2m_{i\alpha}} + V_{iF}^{Coul}, \tag{6}$$

$\mathbf{p}_{i\alpha}$ is the canonical momentum of the $\alpha$ charge of the $i$th molecule, $m_{i\alpha}$ is the corresponding rest mass, and $V_{iF}^{Coul}$ contains the intramolecular longitudinal electrostatic interactions of electrons and nuclei of molecule $i$ in the noninteracting subsensemble of type F. From now on, we consider all molecules involved are nonlinear and polyatomic for the sake of simplicity. Restrictions to the cases where any of the reactants or products are monoatomic, diatomic, or linear molecules can be straightforwardly performed as needed for any application based on the treatment below.

We approximate the light–matter interaction Hamiltonian $H_{LM}$ in the Coulomb gauge as follows:

$$H_{LM} = \sum_{\zeta=1}^{N_M}\left[\mathbf{j}_{\zeta} \cdot \mathbf{A}(\mathbf{X}_{\zeta}) + \bar{e}_{\zeta}^2 \mathbf{A}^2(\mathbf{X}_{\zeta})\right], \tag{7}$$

where $N_M = N_A + N_B + N_C + N_E$, each molecule is labeled by $\zeta$, $\mathbf{X}_{\zeta}$ corresponds to the center of mass of molecule $\zeta$, $e_{\alpha}$ is the charge of particle $\alpha$ (nucleus or electron), $\mathbf{j}_{\zeta} = \sum_{\alpha} e_{\alpha}\mathbf{p}_{\alpha}/m_{\alpha}$ is the charge current operator in the lab (space) frame (with fixed axes) and $\bar{e}_{\zeta}^2 = \sum_{\alpha} e_{\alpha}^2/(2m_{\alpha})$. This Hamiltonian employs a long-wavelength limit approximation that neglects inhomogeneities of the EM field on the spatial scale of the charge density of each molecule. This poses no concern because the infrared field modes of relevance to us have wavelength that is orders of magnitude larger than typical molecular sizes.

A numerical investigation of the partition function associated with the many-body Hamiltonian (Eq. (2)) in terms of its electronic-nuclear-photonic stationary states would require a computationally unfeasible treatment of the intriguing mixed boson-fermion wave functions and the statistics of the light–matter system[41,42]. Therefore, while our description has been general to this point, in what follows we specialize to the case where nonperturbative light–matter inter-actions occur exclusively between high-frequency vibrational normal modes of the molecular ensemble and the electromagnetic modes here considered. We will only be concerned with VSC effects on chemical equilibria at temperatures $T$ that are (a) much greater than

the rotational temperatures $T_{rot}$ of all involved molecules, (b) much lower than the lowest-lying electronic transition of each considered chemical species (at their equilibrium geometry), and (c) much lower than the vibrational temperatures $T_v = h\nu/k_B$ of the bright (dipole-active) normal modes of each molecule (here labeled by the frequency $\nu$). These conditions are generally satisfied by polyatomic molecules at standard conditions of temperature and pressure[1,2], including those employed in VSC experiments that reported polariton generation by the interaction of infrared cavity modes with high-frequency molecular vibrations of polyatomic molecules satisfying $T_v \gg T \gg T_{rot}$[43–47].

Several important implications follow from the conditions (a), (b) and (c) given above. From (a) ($T \gg T_{rot}$), it follows that a classical statistical mechanical treatment of the molecular rotational degrees of freedom provides a reliable approximation to the rotational contribution to the thermodynamic observables of the considered system[1,2]. Therefore, hereafter we consider the rotational degrees of freedom of $h_M$ to be classical phase space variables[48]. Similar considerations can be made for the molecular translational degrees of freedom, for we only examine systems where intermolecular distances are much larger than the thermal de Broglie wavelength of each chemical species. This allows us to treat molecular translational degrees of freedom as classical variables. It follows, in the limit where the light–matter coupling goes to zero, we recover the standard predictions of the statistical mechanics of polyatomic systems, where vibrational modes are treated quantum mechanically and rotations, translations, librations, etc are treated classically[1,2]. As we explain in more detail below, we expect this partitioning of molecular degrees of freedom to be reasonable based on the notion that rotations and translations occur on timescales that are much slower than the vibrational dynamics involved in VSC. Nevertheless, the interaction between fast polaritonic and slow matter degrees of freedom remains an intriguing issue for future work to unravel.

Condition (b) ($k_B T$ is much less than the lowest-lying electronic transition energy of each molecule), implies then that the relevant eigenstates of the noninteracting molecular Hamiltonian $h_{iF}$ are products of the $F$ electronic ground-state and vibrational wave functions. Further simplification arises from condition (c) that $T \ll T_v$ for the molecular normal modes under strong interaction with the EM field. In particular, this condition guarantees negligible thermal populations for states of dipole-active vibrational modes with more than a single quantum (only ground and first excited states are occupied with any significance) at experimentally probed temperatures, so the anharmonicity of modes under VSC is inconsequential for thermal properties of the molecular system, and we can accurately treat the molecular high-frequency modes under VSC as harmonic oscillators without loss of any significant accuracy. Notably, this same argument allows one to treat anharmonic high-frequency modes of molecules as harmonic oscillators and still obtain great agreement with experimental thermodynamic data under standard conditions[1,2].

The fact that vibrational modes with significant oscillator strength satisfy $T_v \gg T$ allows significant simplification of their thermodynamic treatment. In particular, the assumed absence of large-amplitude anharmonic motion implies that the approximate separation of vibrational and rotational motions in the Eckart frame[49–51] is reliable and independent normal-coordinates may be assigned to high-frequency vibrational modes executing small-amplitude motions determined by the lowest vibrational states (for $T \ll T_v$, only the ground and first-excited state associated to each bright normal mode have significant thermal occupation[1,2]). In this case, the molecular infrared polarization operator $\mathbf{P}(\mathbf{x}) = \sum_\zeta \mathbf{P}_\zeta(\mathbf{x})$ can be accurately approximated by retaining only the constant and linear terms in its expansion in powers of the relevant normal-mode displacements (those associated with transitions with significant oscillator strength)

$(\mathbf{Y}_{1\zeta}, \mathbf{Y}_{2\zeta}, ..., \mathbf{Y}_{n_\zeta, \zeta})$[52], where each of the normal displacements is written in the molecular frame (i.e., that rotates with the molecular system), $n_\zeta$ is the number of $\zeta$ normal modes with significant oscillator strength ($n_\zeta \le 3N_{a\zeta} - 6$ for each molecule $\zeta$ with $3N_{a\zeta}$ atoms and $3N_{a\zeta} - 6$ normal modes in its electronic ground-state, as we have for simplicity assumed all involved molecules are nonlinear). Using the electrical dipole approximation to the molecular polarization operator[38], it follows that $\mathbf{P}_\zeta(\mathbf{x})$ can be written as

$$\mathbf{P}_\zeta(\mathbf{x}) = R_\zeta^T(\boldsymbol{\theta}_\zeta)\left[\mathbf{d}_{0\zeta} + \Delta\mathbf{d}_\zeta(\mathbf{Y}_\zeta)\right]\delta(\mathbf{x} - \mathbf{X}_\zeta), \tag{8}$$

$$\Delta\mathbf{d}_\zeta(\mathbf{Y}_\zeta) = \sum_{i=1}^{n_\zeta}(\mathbf{Y}_{i\zeta} \cdot \nabla_Y)\mathbf{d}_\zeta(\mathbf{Y})|_{\mathbf{Y}=0}, \tag{9}$$

where $\boldsymbol{\theta}_\zeta$ denotes the Euler angles specifying the orientation of molecule $\zeta$, $R_\zeta^T(\boldsymbol{\theta}_\zeta)$ is the SO(3) rotation that maps the $\zeta$ frame into the lab frame[48], $\mathbf{d}_{0\zeta}$ is the molecular dipole moment operator at its equilibrium geometry ($\mathbf{Y} = 0$) in the molecular frame, and $\mathbf{Y}_{i\zeta} \cdot \nabla_Y$ is the corresponding nuclear gradient along the normal-mode displacement $\mathbf{Y}_{i\zeta}$.

The transition-matrix elements associated to the molecular current operators $\mathbf{j}_\zeta$ can be related to the matter polarization contribution from each molecule using the identity $\mathbf{j}_\zeta = \sum_\alpha e_\alpha \mathbf{p}_{\zeta\alpha}/m_{\zeta\alpha} = -i\sum_\alpha e_\alpha[x_{\zeta\alpha}, h_\zeta]/\hbar$. In the basis of vibrational eigenstates of $h_\zeta$ with corresponding eigenvalues $E_{\zeta a}$, Equations (8) and (9) imply $\mathbf{j}_\zeta^{ab} = R_\zeta^T(\boldsymbol{\theta}_\zeta)[i\omega_\zeta^{ab}\mathbf{d}_\zeta^{ab}(\mathbf{Y}_\zeta)]$, where $\omega_\zeta^{ab} = (E_{\zeta a} - E_{\zeta b})/\hbar$. The contribution of each molecule to the diamagnetic term of $H_{LM}$ (Eq. (7)), namely $\bar{e}_\zeta^2\mathbf{A}^2(\mathbf{X}_\zeta)$ must also be reconsidered in light of the restriction of our nonperturbative treatment to infrared molecular transitions, e.g., the Thomas-Reiche-Kuhn sum rule[53–55] can be employed to obtain in the restricted molecular Hilbert space a diamagnetic term consistent with the approximations here employed to model VSC.

The described approach, where high-frequency vibrations and EM modes are treated quantum mechanically, electronic degrees of freedom are frozen in the ground-state, and translations and rotations are treated classically leads to a family of positive-definite quadratic Hamiltonians $h = h_M(\mathbf{J}, \mathbf{P}_C) + H_L + H_{LM}(\boldsymbol{\theta}, \mathbf{X})$ parametrized by the set of center of mass position and momentum of each molecule $(\mathbf{X}_1, \mathbf{P}_{C_1}, ..., \mathbf{X}_{N_M}, \mathbf{P}_{C_{N_M}}) = (\mathbf{X}, \mathbf{P}_C)$ and their corresponding (classical) orientations and angular momenta $(\boldsymbol{\theta}_1, \mathbf{J}_1, ..., \boldsymbol{\theta}_{N_M}, \mathbf{J}_{N_M}) = (\boldsymbol{\theta}, \mathbf{J})$. Each set of classical molecular variables leads to a Hamiltonian $h$ with two classes of eigenmodes: (i) polaritons with frequency $\omega_{\mathcal{P}}(\boldsymbol{\theta}, \mathbf{X}) > 0$ (these could include modes with negligible fraction of EM or molecular excitation) and (ii) dipole-inactive vibrations described by the Hamiltonian $H_D$ which we define such that it also includes the ground-state electronic energy and classical translational and rotational kinetic energy of each molecule. We infer (under the separability conditions and classical correspondence previously delineated), the thermodynamic properties of the total light–matter Hamiltonian (Eq. (2)) can be obtained from the statistical mechanical treatment of the effective Hamiltonian (here written on the basis of eigenmodes of $h_M(\mathbf{J}, \mathbf{P}_C) + H_L + H_{LM}(\boldsymbol{\theta}, \mathbf{X})$),

$$H(\mathbf{J}, \boldsymbol{\theta}, \mathbf{P}_C, \mathbf{X}) = H_{Pol}(\boldsymbol{\theta}, \mathbf{X}) + H_D(\mathbf{J}, \mathbf{P}_C) + V_M(\boldsymbol{\theta}, \mathbf{X}), \tag{10}$$

where $H_{Pol}(\boldsymbol{\theta}, \mathbf{X})$ represents the (in general, multimode) vibrational polariton Hamiltonian in the normal-mode representation, $H_D$ is the sum of (classical) molecular translational and rotational kinetic energies and dipole-inactive (normal-mode quantum) vibrational Hamiltonians, and $V_M(\boldsymbol{\theta}, \mathbf{X})$ describes intermolecular interactions. Note the exclusion of nonadiabatic terms in Eq. (10) is consistent with the assumed separability of fast and slow molecular degrees of freedom. While nonadiabatic interactions drive relaxation and are potentially key ingredients in dynamics, we leave for future work to precisely discern

their relevance for the equilibrium statistical mechanics of polaritonic materials.

The polaritonic part of the Hamiltonian can be expressed in terms of polariton creation and annihilation operators as follows:

$$H_{\text{Pol}}(\boldsymbol{\theta}, \mathbf{X}) = \sum_{\mathcal{P}} \hbar\omega_{\mathcal{P}}(\boldsymbol{\theta}, \mathbf{X})\left(c_{\mathcal{P}}^{\dagger}c_{\mathcal{P}} + \tfrac{1}{2}\right), \tag{11}$$

where $c_{\mathcal{P}}^{\dagger}$ $(c_{\mathcal{P}})$ is the bosonic creation (annihilation) operator associated with polariton mode $\mathcal{P}$. Note that $H_{\text{Pol}}, c_{\mathcal{P}}^{\dagger}, c_{\mathcal{P}}$, and $\omega_{\mathcal{P}}$ are all dependent on the molecular orientations $\boldsymbol{\theta}_{\zeta}$, center of mass positions $\mathbf{X}_{\zeta}$ and number of molecules of each chemical species contributing to the formation of the polariton modes $\mathcal{P}$. These eigenmodes may be more or less localized depending on their energy and the typical size of fluctuations of the molecular ensemble (disorder)[56–58].

Intermolecular interactions represented by $V_M$ include nonlinear couplings between degrees of freedom with free dynamics generated by both $H_{\text{D}}$ and $H_{\text{Pol}}$. These interactions induce polariton decay and contribute to their linewidths[59,60]. For the sake of simplicity, we will proceed with the assumption that the intermolecular interactions modeled by $V_M$ can be ignored for the purpose of computing the thermodynamic equilibrium properties of the molecular system. This assumption can and will be lifted later. Therefore, we employ

$$h(\mathbf{J}, \boldsymbol{\theta}, \mathbf{X}, \mathbf{P}_C) \equiv H_{\text{Pol}}(\boldsymbol{\theta}, \mathbf{X}) + H_{\text{D}}(\mathbf{J}, \mathbf{P}_C) \tag{12}$$

to compute the canonical ensemble partition function of the light−matter system and its corresponding thermal properties.

The partition function of the light−matter system at fixed volume $V$, temperature $T$, and $N_A, N_B, N_C, N_E$ molecules can be written as

$$Q(N, V, T) = \int \frac{\mathrm{d}\mathbf{J}\mathrm{d}\boldsymbol{\theta}\mathrm{d}\mathbf{X}\mathrm{d}\mathbf{P}_C}{(2\pi\hbar)^{12N_M}} \text{Tr}\left[e^{-\beta h(\mathbf{J}, \boldsymbol{\theta}, \mathbf{X}, \mathbf{P}_C)}\right] \tag{13}$$

where $N = (N_A, N_B, N_C, N_E)$, $\mathrm{d}\mathbf{J}\mathrm{d}\boldsymbol{\theta}\mathrm{d}\mathbf{X}\mathrm{d}\mathbf{P}_C$ is a compact notation for the $12N$-symplectic form of the translational-rotational molecular phase space, and Tr denotes the trace over quantum states of all light and matter (vibrational) degrees of freedom. The integration over $\mathbf{J}$ and $\mathbf{P}_C$ is trivial, for the Hamiltonian is quadratic in these variables. Further, we make the approximation that the molecular orientational and positional distributions are isotropic and uniform, respectively and unaffected by the interaction with the considered EM modes. These considerations imply that under the collective light−matter interaction regime, the molecular rotational and translational partition functions are unaffected by VSC, and the total partition function of the light−matter system can be approximated as

$$Q(N, V, T) = Q_{\text{D}}(N, V, T) Q_{\text{Pol}}(N, V, T), \tag{14}$$

where $Q_{\text{D}}(N, V, T)$ denotes the partition function for the electronic-translational-rotational and non-dipole active vibrational degrees of freedom

$$Q_{\text{D}}(N, V, T) = \prod_{F \in \{A, B, C, E\}} \frac{\left[q_{\text{el}}^{F}(T)q_{\text{trans}}^{F}(V, T)q_{\text{rot}}^{F}(T)\tilde{q}_{\text{vib}}^{F}(T)\right]^{N_F}}{N_F!}, \tag{15}$$

where $\tilde{q}_{\text{vib}}^{F}(T)$ is the partition function associated with the dipole-inactive normal modes of a single F molecule.

The polariton partition function $Q_{\text{Pol}}(N, V, T)$ is given by the macroscopic average of $Q_{\mathcal{P}}(\boldsymbol{\theta}, \mathbf{X}) = \text{Tr}\left[\exp(-\beta H_{\text{Pol}}[\boldsymbol{\theta}, \mathbf{X}])\right]$ over the space of molecular positions $\mathbf{X}$ and orientations $\boldsymbol{\theta}$. Assuming the

molecular system is isotropic and uniformly distributed (over long distances), it follows that

$$\begin{aligned} Q_{\text{Pol}}(N, V, T) &= \int \frac{\mathrm{d}\boldsymbol{\theta}\mathrm{d}\mathbf{X}}{(2\pi\hbar)^{6N_M}} Q_{\mathcal{P}}(\boldsymbol{\theta}, \mathbf{X}) \\ &= \int \frac{\mathrm{d}\boldsymbol{\theta}\mathrm{d}\mathbf{X}}{(2\pi\hbar)^{6N_M}} \text{Tr} \exp\left[-\beta H_{\text{Pol}}(\boldsymbol{\theta}, \mathbf{X})\right]. \end{aligned} \tag{16}$$

The Helmholtz free energy of the light−matter system can now be directly obtained from $Q(N, V, T)$ as

$$\begin{aligned} A(N, V, T) &= -\mathrm{k_B}T \ln\left[Q(N, V, T)\right], \\ &= A_{\text{Pol}}(N, V, T) + A_{\text{D}}(N, V, T), \end{aligned} \tag{17}$$

where $A_{\text{Pol}}(N, V, T) = -\mathrm{k_B}T \ln\left[Q_{\text{Pol}}(N, V, T)\right]$ (with $Q_{\text{Pol}}(N, V, T)$ corresponding to the macroscopically averaged polariton contribution to $Q(N, V, T)$ as given by Eq. (16)). Likewise applies for $A_{\text{D}}(N, V, T)$, which is the free energy of the modes with dynamics generated by $H_{\text{D}}$.

The chemical equilibrium condition at fixed $V$ and $T$ is[1,2]

$$\begin{aligned} \sum_{\text{F}} \tilde{\nu}_{\text{F}} \frac{\partial A}{\partial N_{\text{F}}} &= \tilde{\nu}_{\text{A}}\mu_{\text{A}} + \tilde{\nu}_{\text{B}}\mu_{\text{B}} + \tilde{\nu}_{\text{C}}\mu_{\text{C}} + \tilde{\nu}_{\text{E}}\mu_{\text{E}} \\ &= 0, \end{aligned} \tag{18}$$

where $\tilde{\nu}_{\text{F}} = \nu_{\text{F}}$ if F is a product species and $\tilde{\nu}_{\text{F}} = -\nu_{\text{F}}$ is a reactant, and the chemical potentials $\mu_{\text{F}}(N, V, T) = \partial A/\partial N_{\text{F}}$ are given by

$$\mu_{\text{F}}(N, V, T) = \mu_{\text{F, Pol}}(N, V, T) + \mu_{\text{F, D}}(N_F, V, T), \tag{19}$$

with $\mu_{\text{F,Pol}}(N, V, T) = \partial A_{\text{Pol}}/\partial N_{\text{F}}$ and $\mu_{\text{F,D}}(N_F, V, T) = \partial A_{\text{D}}/\partial N_{\text{F}}$. Note that $\mu_{\text{F,Pol}}(N, V, T)$ corresponds to the change in the chemical potential of species F induced by the strong light−matter interaction and is unrelated to the polaritonic chemical potential. This quantity vanishes at thermal equilibrium as follows for any non-conserved quasiparticles[2,61].

We can rewrite Eq. (19) in terms of a bare contribution and a polariton-induced change by adding and subtracting the contribution to the chemical potential from the bright vibrational part of $h_F$ which we write as $\mu_{\text{F,vib,bright}}(T)$. Given that $\mu_{\text{F,vib,bright}}(T)$ is the contribution of bright vibrational modes to the bare chemical potential for a system of noninteracting $N_F$ molecules of type F, and the remaining additive contribution to the F chemical potential is $\mu_{\text{F,D}}$, we define the reference chemical potential of species F by

$$\mu_{\text{F}}^{(0)}(N_F, V, T) = \mu_{\text{F, D}}(N_F, V, T) + \mu_{\text{F, vib, bright}}(T). \tag{20}$$

Note that $\mu_{\text{F}}^{(0)}(N_F, V, T)$ is employed as a standard-state relative to which the molecular chemical potential is obtained under conditions where interaction of the molecular system with the electromagnetic field may be significant. Equivalently, $\mu_{\text{F}}^{(0)}$ follows from the same separability conditions employed to obtain Eq. (14) in the limit where the light−matter interaction approaches zero. It follows that the chemical potential of species F under the influence of the EM field is given by

$$\mu_{\text{F}}(N, V, T) = \mu_{\text{F}}^{(0)}(N_F, V, T) + \Delta\mu_{\text{F, Pol}}(N, V, T), \tag{21}$$

where we introduced

$$\Delta\mu_{\text{F, Pol}} = \mu_{\text{F, Pol}} - \mu_{\text{F, vib, bright}}. \tag{22}$$

The bare F chemical potential can be obtained directly from Eqs. (20) and (15) as

$$\mu_{\mathrm{F}}^{(0)}(N_F, V, T) = -k_B T \frac{\partial}{\partial N_F} \ln\left[\frac{q_{\mathrm{F}}^{N_F}(V, T)}{N_F!}\right]$$
$$= -k_B T \ln\left[\frac{q_{\mathrm{F}}(V, T)}{N_F}\right], \quad N_F \gg 1. \tag{23}$$

where $q_{\mathrm{F}}(V, T) = q_{\mathrm{el}}^{\mathrm{F}}(T) q_{\mathrm{trans}}^{\mathrm{F}}(V, T) q_{\mathrm{rot}}^{\mathrm{F}}(T) q_{\mathrm{vib}}^{\mathrm{F}}(T)$ is the single-molecule partition function of the bare isolated species F, and to obtain the second line we employed Stirling's approximation. By applying Eqs. (23) and (21) into Eq. (18), we obtain

$$-k_B T \sum_F \tilde{\nu}_F \left[\frac{q_{\mathrm{F}}}{N_F}\right] + \sum_F \tilde{\nu}_F \Delta\mu_{\mathrm{F,Pol}} = 0. \tag{24}$$

A simple rearrangement leads to our expression for the equilibrium reaction quotient (Eq. (25)) under the influence of nonperturbative light–matter interactions

$$\frac{N_{\mathrm{E}}^{\nu_E} N_{\mathrm{C}}^{\nu_C}}{N_{\mathrm{A}}^{\nu_A} N_{\mathrm{B}}^{\nu_B}} = \frac{q_{\mathrm{E}}^{\nu_E} q_{\mathrm{C}}^{\nu_C}}{q_{\mathrm{A}}^{\nu_A} q_{\mathrm{B}}^{\nu_B}} \exp\left[-\beta \sum_F \tilde{\nu}_F \Delta\mu_{\mathrm{F,Pol}}(N_A, N_B, N_C, N_E, V, T)\right]. \tag{25}$$

By solving Eq. (25) for the number of molecules of each species under constraints set by the experimental situation (e.g., the system is initially prepared with an equal number of A and B molecules, etc), we obtain the polariton effect on the equilibrium composition of the reactive mixture.

## Qualitative analysis of VSC effects on chemical equilibria

Generic properties of vibrational polariton effects on chemical equilibria arising as a consequence of Eq. (25) are summarized here. First, Eq. (25) shows a proper description of polariton effects on chemical equilibria requires a multimode description of the EM field, as both on and off-resonant modes contribute to the VSC-induced changes in the reaction quotient. Explicitly, in a system with $N_P$ eigenmodes (Eq. (11)), we can take advantage of the assumed condition that $T_\nu \gg T$ (for modes strongly coupled to light) and the consequent quadratic nature of the strong light–matter system to obtain $Q_{\mathrm{Pol}} = \prod_{l=1}^{N_P} q_{P_l}$, where $q_{P_l}$ is the harmonic oscillator partition function associated to the $l$th polariton mode. It follows the polariton contribution to the free energy is additive, with $A_{\mathrm{Pol}} = \sum_{l=1}^{N_P} A_{P_l}$, where $A_{P_l} = -k_B T \ln(q_{P_l})$. Hence, the polariton-induced change in the matter chemical potential $\Delta\mu_{\mathrm{F,Pol}}$ has contributions from all $N_P$ polariton modes with participation of chemical species F. In fact, using $\Delta\mu_{\mathrm{F,Pol}} = \sum_{l=1}^{N_P} \mu_{F,P_l} - \mu_{\mathrm{F,vib,bright}}$, where $\mu_{F,P_l} = \partial_{N_F} A_{P_l}$, we find directly that the r.h.s of Eq. (25) depends on all $N_P$ polariton modes formed via hybridization with any of the molecular species. Clearly, no a priori special role is played by field fluctuations corresponding to incidence angles near zero, and devices with greater density of polariton modes will allow greater control of chemical equilibria. Additionally, it is seen that polaritons originating from all bands of a microcavity in resonance or sufficiently close to resonance with dipole-active molecular vibrations will contribute to Eq. (25).

Second, Eq. (25) demonstrates that in a polyatomic system with multiple bright vibrations, the chemical equilibrium shift induced by an IR microcavity depends on the density of EM modes at the various bright IR resonances of both reactants and products and their corresponding oscillator strengths. For instance, if the chemical species F has $n_F$ bright normal modes in resonance or near-resonance with EM modes corresponding to any incidence angle, then $\mu_{\mathrm{F,vib,bright}}$ has additive contributions from all bright normal modes $m = 1, 2, \ldots, n_F$, i.e, $\mu_{\mathrm{F, vib, bright}} = \sum_m \mu_F^{(m)}$, where $\mu_F^{(m)}$ is the chemical potential associated with the $m$th vibrational mode of each molecule, and $\Delta\mu_{\mathrm{F,Pol}}$ will

be impacted by all such polaritons formed between the $F$ chemical species and the confined EM field.

Equation (25) also indicates that VSC may shift the equilibrium towards products or reactants depending only on their oscillator strength density, vibrational resonance frequencies, and the spectrum of the confined EM field. These quantities control the polariton contributions to the chemical potential $\Delta\mu_{\mathrm{F,Pol}}$ via its dependence on the polariton energies, as these are determined by the collective interaction strengths of the various bright vibrational modes involved in a typical equilibrium. In fact, VSC will lead to a greater fraction of product species, when the polariton-induced change in the chemical potential of the products $\Delta\mu_{\mathrm{P,Pol}} = \nu_C \Delta\mu_{\mathrm{C,Pol}} + \nu_D \Delta\mu_{\mathrm{D,Pol}}$ is less than the corresponding quantity for the reactants $\Delta\mu_{\mathrm{R,Pol}} = \nu_A \Delta\mu_{\mathrm{A,Pol}} + \nu_B \Delta\mu_{\mathrm{B,Pol}}$, for in this case $\exp[-\beta(\Delta\mu_{\mathrm{P,Pol}} - \Delta\mu_{\mathrm{R,Pol}})]$ is greater than one, so it follows from Eq. (25) that the reaction quotient under VSC as expressed by $N_C^{\nu_C} N_E^{\nu_E} / [N_A^{\nu_A} N_B^{\nu_B}]$ is greater than the reaction quotient in free space $[q_C^{\nu_C} q_E^{\nu_E} / (q_A^{\nu_A} q_B^{\nu_B})]$. In the next section, we quantitatively investigate $\Delta\mu_{\mathrm{F,Pol}}$ in an elementary model of strong light–matter coupling to confirm the validity of these statements.

Our results relied primarily on conditions accessed by the vast majority of VSC experiments (moderate temperatures that are much smaller than all electronic transition energies and vibrational temperatures of high-frequency modes contributing to polaritons, but also much greater than rotational temperatures and isotropic molecular orientation distribution negligibly perturbed by interaction with both polarizations of the EM field). We ignored the intermolecular term $V_M$ and anharmonicity even in low-frequency vibrational modes (assumed to be weakly coupled to the radiation field) to obtain Eq. (25), but these approximations can be easily made much less extreme without almost any change in our formalism. For instance, we can add intramolecular anharmonicity to the low-frequency modes without any change to Eq. (25) by employing an anharmonic vibrational partition for $\tilde{q}_{\mathrm{vib}}(T)$. This could include nonlinear couplings between modes that are not involved in polariton formation, or anharmonic interactions that only significantly perturb highly-excited polariton modes (with at least $\nu \geq 2$) with negligible thermal occupation at the considered temperatures $T \ll T_\nu$. Likewise, we can reintroduce without any additional complexity, the effects of the intermolecular longitudinal electrostatic interactions $V_M$ on the rotational, translational and vibrational modes with weak or vanishing oscillator strength. This procedure would lead to a new standard state for the reaction quotient outside a microcavity, i.e., $q_E^{\nu_E} q_C^{\nu_C} / (q_A^{\nu_A} q_B^{\nu_B})$ would be converted into the expression of the equilibrium reaction quotient in free space accounting for the considered longitudinal interactions between all present chemical species.

Changes in the longitudinal EM interactions induced by any nontrivial boundary conditions satisfied by the EM field[39] could be accounted for by writing $V_M = V_M^0 + \Delta V_M$, where $V_M^0$ is the free-space electrostatic potential and $\Delta V_M$ accounts for the renormalization of the free-space Coulomb potential[37]. Explicit inclusion of this term would lead to another contribution to the field-matter change in the chemical potential of each species in the reactive mixture. Note that our main result makes no simplification nor assumption about the existence of energetic disorder which may weakly perturb normal-mode frequencies and change the equilibrium reaction quotient (Eq. (25)) via the disorder-induced variation of $\Delta\mu_{\mathrm{F,Pol}}$. In the simplest case where molecular interactions with an inert background lead to static disorder corresponding to small fluctuations in normal-mode frequencies, renormalized thermal observables could be obtained from the partition function resulting from the disorder-average of $Q_{\mathrm{Pol}}(N, V, T; \xi)$ obtained at a particular disorder realization $\xi$. The same procedures that led to Eq. (25) would apply with (disorder-averaged) renormalized quantities.

To conclude this discussion of our formalism, we note that, under collective vibrational strong coupling, polariton frequencies $\omega_{\mathcal{P}}$ depend on the orientation and center of mass coordinates of a large number of molecules, and therefore, we expect negligible fluctuations in the spectrum of $H_{\text{Pol}}(\boldsymbol{\theta}, \mathbf{X})$ from its macroscopic average. This feature suggests a simple approximation to the light–matter partition function

$$
\begin{aligned}
Q_{\text{Pol}}(N, V, T) &\approx \prod_{\mathcal{P}} \bar{q}_{\mathcal{P}}(T) \\
&= \prod_{\mathcal{P}} \frac{e^{-\beta \hbar \bar{\omega}_{\mathcal{P}}/2}}{1 - e^{-\beta \hbar \bar{\omega}_{\mathcal{P}}}},
\end{aligned}
\tag{26}
$$

where $\bar{q}_{\mathcal{P}}$ and the corresponding frequencies $\bar{\omega}_{\mathcal{P}}$ are harmonic partition functions and frequencies obtained from the (uniform and isotropic) translational-orientational average of the normal-mode spectrum of the quadratic polariton Hamiltonian (Eq. (11)). Several methods can be employed to estimate the mean frequencies $\bar{\omega}_{\mathcal{P}}$[62–65]. For example, in their study of polariton scattering and localization, refs. 62, [63] obtained macroscopically averaged polariton frequencies in the rotating-wave-approximation and the same methods can be applied to generate mean-field normal-mode frequencies of any positive-definite quadratic light–matter Hamiltonian.

## Application to single-mode cavity strongly coupled to reactant subensemble

In the next section, we apply our theory to a reactive mixture where a single subensemble of a molecular system strongly interacts with a microcavity represented by a single boson mode. This is a highly idealized scenario relative to most experiments for the reasons that we indicated above, e.g., polyatomic molecules have multiple bright vibrational modes and a continuous set of on and off-resonant EM modes contribute to polariton effects on chemical equilibria in planar microcavities. We also ignore disorder effects by assuming a 0D microcavity geometry and a perfectly oriented molecular ensemble. This limit is equivalent to assuming trivial probability distributions (Dirac delta functions) for $\mathbf{X}_\zeta, \boldsymbol{\theta}_\zeta$ and the matter normal-mode frequencies.

As we demonstrate below, while we invoke idealized conditions, our analysis of polaritonic effects on equilibria in single-mode EM resonators already indicates several important qualitative trends that are expected to persist in any complete treatment including a macroscopic number of molecular and EM degrees of freedom.

In the case where only reactant species A strongly interacts with a single EM mode and the number of molecules of type A (obtained from solving Eq. (25)) is $N_A$, the nonperturbative light–matter Hamiltonian contains $N_A + 1$ eigenmodes corresponding to the $N_A - 1$ purely molecular modes that have the same spectrum as the bright vibrations of A and the hybrid LP and UP. The contribution of the $N_A - 1$ reservoir normal modes to $\mu_{\text{F,Pol}}$ cancels out the term $\mu_{A,\text{vib,bright}}(T)$ in $\Delta\mu_{A,\text{Pol}}(N, V, T)$ (Eq. (22)). As expected, the effect of nonperturbative light–matter interactions on the composition of the reactive mixture at equilibrium in this example is entirely due to the isolated LP and UP modes. Let the polariton effect on chemical equilibrium $F_{\text{Pol}}(V, T)$ be defined as the ratio of the equilibrium reaction quotient inside the microcavity $R(V, T) = N_E^{\nu_E} N_C^{\nu_C} / N_A^{\nu_A} N_B^{\nu_B}$ to the standard-state reaction quotient (equilibrium constant) $K^{(0)}(T) = (q_E^{\nu_E} q_C^{\nu_C})/(q_A^{\nu_A} q_B^{\nu_B})$ (assuming ideal-gas conditions for simplicity). It follows from Eq. (25) that at equilibrium the polariton effect on the reaction quotient is given by

$$
\begin{aligned}
F_{\text{Pol}}(V, T) &= \frac{R(V, T)}{K^{(0)}(T)} \\
&= e^{\beta\nu_A[\mu_A^{\text{LP}}(N_A, V, T) + \mu_A^{\text{UP}}(N_A, V, T)]},
\end{aligned}
\tag{27}
$$

where $N_A$ is obtained by solving the equation $R(V, T) = K^{(0)}(T) e^{\beta\nu_A[\mu_A^{\text{LP}}(N_A, V, T) + \mu_A^{\text{UP}}(N_A, V, T)]}$, and the changes in the chemical

**Table 1 | Selected IR-active vibrational modes of C₂H₅Cl (P modes) and C₂H₅Br (R modes) with ω obtained from ref. 80 and vibrational temperatures $T_v = h\nu/k_B$, where $\nu$ is the frequency of each mode**

| P mode | $\omega$(cm⁻¹) | $T_v$(K) | R mode | $\omega$(cm⁻¹) | $T_v$(K) |
|---|---|---|---|---|---|
| CC Str | 974 | 1402 | CC Str | 964 | 1388 |
| CCl Str | 677 | 974 | CBr Str | 583 | 839 |

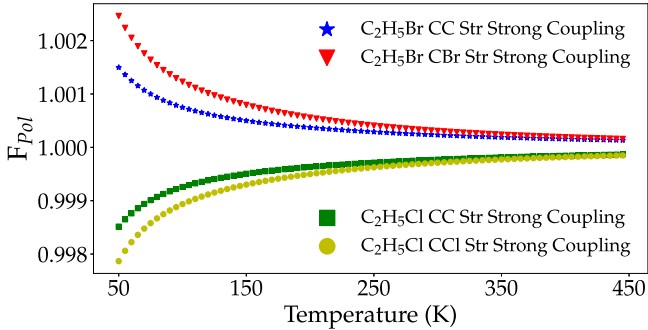

**Fig. 2 | Temperature dependence of (single-mode) polariton effect on the examined S$_N$2 equilibrium.** Computed (single-mode) strong light–matter interaction effect on the examined reactive equilibrium ($F_{\text{Pol}}^R$ in the case of reactant vibrational strong coupling (VSC) and $F_{\text{Pol}}^P$ for product VSC as obtained from Eqs. (40) and (41), respectively) as a function of temperature $T$ (K). Each curve corresponds to a scenario with exclusive VSC between the denoted normal mode and a corresponding resonant EM mode. The single-molecule light–matter coupling strength is $g = 10$ cm⁻¹ and the maximum number of strongly coupled modes is $N = 100$. These results show polariton formation shifts the equilibrium towards reactants in the case where products are strongly coupled and vice-versa. The magnitude of the effect is seen to be inversely correlated to the natural frequencies of the strongly coupled molecular modes (Table 1), i.e., assuming equal light–matter coupling strength, modes with lower frequencies lead to greater polariton-induced changes in equilibrium reaction quotient as measured by $|F_{\text{Pol}} - 1|$.

potential of the A subensemble due to LP and UP are given by

$$
\mu_A^{\text{LP}}(N_A, V, T) = \frac{\partial A_{\text{LP}}(N_A, V, T)}{\partial N_A},
\tag{28}
$$

where $A_{\text{LP}} = -k_B T \ln q_{\text{LP}}(N_A, V, T)$, and identical definitions exist for UP. Equation 27 forms the basis for the qualitative and quantitative analysis of a model gas-phase bimolecular nucleophilic substitution reaction that we discuss in the next section.

Note that, as is well known[58,66,67], the degeneracy of the $N_A - 1$ dark modes is easily broken as they become weakly coupled to light in the presence of molecular permutational-symmetry breaking perturbations. This does not change Eq. (27) in any appreciable way, since the difference between the weakly coupled reservoir density of states and that of the molecular system in free space is negligible in the collective strong light–matter interaction regime of interest to us[58,68]. Therefore, the same cancellation between the free space bright vibrational contribution to the chemical potential of the A subensemble and the molecular dark reservoir inside an optical cavity occurs to a large extent when the number of molecules is sufficiently large, i.e., when $N_A \to \infty$, and thus the results obtained in the presence of permutational symmetry $\Delta\mu_{A, \text{Pol}} = \mu_A^{\text{LP}} + \mu_A^{\text{LP}}$ remain a very good approximation for a molecular ensemble interacting with a single boson mode.

## Bimolecular nucleophilic substitution equilibrium model in a single-mode cavity

To illustrate the theory described above, we consider a lossless single-mode cavity interacting with a gas-phase reactive mixture where

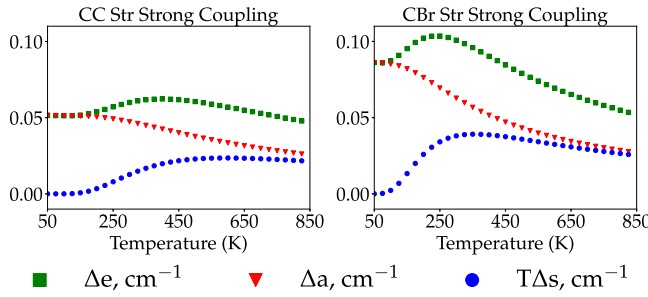

Fig. 3 | Polariton-induced changes in thermodynamic observables of the light–matter system.

**Fig. 3 | Polariton-induced changes in thermodynamic observables of the light–matter system.** Temperature ($T$) dependence of changes promoted by vibrational strong coupling (VSC) on the free energy ($\Delta a$, Eq. (31)), internal energy ($\Delta e$, Eq. (30)) and entropic contribution to the free energy per unit interacting degree of freedom ($T\Delta s$, Eq. (32)) of the light–matter system. The left figure shows results obtained for the case where the CC Str mode of $C_2H_5Br$ is strongly coupled to the single resonant EM mode, with (single-molecule) light–matter coupling strength $g = 10$ cm$^{-1}$ and total number of reactants and product molecules $N = 100$. The right figure shows analogous results when the reactant CBr Str vibration is strongly coupled to the single-mode electromagnetic field. These results reveal that polariton-induced changes in the system's zero-point energy provide the dominant contribution to the temperature-dependent single-mode VSC effect on chemical equilibrium reported in Fig. 2, and that entropic changes become relevant as $T$ grows from ultracold ($T < 100$ K) to the vibrational temperature $h\nu/k_B$ of the strongly coupled mode with frequency $\nu$. The change in internal energy per molecule is an increasing function of the temperature at low $T$ because the lower polariton (LP) mode frequency is lower than the bare molecular, and thus LP has greater likelihood to thermally occupy the state with a single quantum than the bare molecular vibration at $T \ll T_\nu$. This same variation of occupation number explains the observed polariton-induced increase in the total entropy at low $T$. The computed effects are seen to be small, but the quantitative details apply strictly only to the case where a single cavity mode interacts with a single normal mode of the reactant or product ensemble.

equilibrium is established via the $S_N2$ reaction

$$C_2H_5Br + Cl^- \rightleftharpoons C_2H_5Cl + Br^-. \qquad (29)$$

This reaction has been thoroughly studied in the gas phase[69,70]. We construct its chemical equilibrium constant in free space from the gas-phase partition function of each chemical species assuming separability between the internal degrees of freedom and ideal gas conditions.

In order to probe polariton effects on the chemical equilibrium associated with Eq. (29), we suppose the system is embedded in an optical cavity with a single high-quality mode in resonance with a particular vibrational mode of reactants or products. To examine the distinct effects of reactant and product strong light–matter coupling, we chose two strongly absorbing IR modes of reactants and products[71]. The frequencies and vibrational temperatures of the selected dipole-active normal modes are given in Table 1.

**Temperature, coupling strength and system-size dependence of single-mode polariton effects on a model equilibrium**

We have investigated the effect of single-mode strong light–matter coupling on the equilibrium composition of the reactive molecular mixture described by Eq. (29) at various temperatures, system sizes, and light–matter interaction strengths assuming that strong coupling occurs between the cavity and a single set of normal vibrational modes of reactant or product.

The bare cavity frequency $\omega_C$ is set to be in resonance with the strongly coupled vibrational mode. Note renormalization (see Methods) of the cavity frequency in the presence of the molecular system leads to a nonzero detuning that is insignificant relative to the

light–matter interaction strength under the conditions analyzed in this work.

The temperature dependence of the ratio between the reaction quotient of the selected $S_N2$ reaction inside and outside a microcavity is provided in Fig. 2. This figure shows four notable features: a. polariton effects are strongest at low temperatures and vanish at the high-temperature limit, b. the equilibrium is shifted towards the products ($C_2H_5Cl + Br^-$) when reactants are strongly coupled to light and vice-versa, c. the computed effects are especially negligible considering the large single-molecule light–matter coupling strength employed (for the purposes of illustrating our theory), and d. polaritons formed between molecular modes with lower frequency have a stronger impact on the chemical equilibrium. Below, we discuss each of these trends.

**Low- and high-temperature behavior.** Figure 2 shows the single-mode cavity effect on the composition of the equilibrium reactive mixture is largest at low temperatures, whereas strong coupling has no effect in the high-temperature limit. To understand this, note that at low temperatures, the polaritons and bare modes are essentially in their ground-state, and therefore any polariton-induced change in free energy responsible for modifying chemical equilibrium is generated by the difference between polariton and bare molecule zero-point energies. At high temperatures, the classical limit of the light–matter partition function can be employed to show that the free energy of the reactive mixture is unaffected by polariton formation[35,36].

In Fig. 3, we examine the polariton-induced variation with temperature of the change per strongly coupled degree of freedom (molecular and photonic) in the internal energy $\Delta e = \Delta E/(N_F + 1)$, free energy $\Delta a = \Delta A/(N_F + 1)$ and $T\Delta s = T\Delta S/(N_F + 1)$ of the system at equilibrium

$$\Delta e = \frac{E_{LP} + E_{UP} - E_F - E_C}{N_F + 1}, \qquad (30)$$

$$\Delta a = \frac{A_{LP} + A_{UP} - A_F - A_C}{N_F + 1}, \qquad (31)$$

$$T\Delta s = \frac{S_{LP} + S_{UP} - S_F - S_C}{N_F + 1}, \qquad (32)$$

where F is either R or P and $N_F$ is the number of strongly coupled molecules at equilibrium. We limit our discussion to strong coupling with the reactant ensemble ($C_2H_5Br$) since the conclusions we derive here are straightforwardly generalizable to the case where strong coupling occurs with products.

Figure 3 shows the observed polariton effect in the reactive mixture composition (Fig. 2) at low $T$ is essentially due to the cavity-induced change in reactant or product zero-point energies. This follows from the fact that at the low-$T$ limit, $\Delta e$ is entirely determined by the zero-point energy of the degrees of freedom involved in strong light–matter coupling

$$\lim_{T \to 0} \Delta e = \frac{\hbar(\omega_{LP} + \omega_{UP} - \omega_F - \omega_C)}{2(N_F + 1)}. \qquad (33)$$

Conversely, the entropy contribution of all modes vanish as $T \to 0$. Therefore, it follows, given that $\omega_{LP} + \omega_{UP} - \omega_F - \omega_C \neq 0$, the change in system free energy induced by the optical cavity at low temperatures relative to the vibrational temperature of the strongly coupled modes is dominated by the ground-state energy difference between the polariton normal modes and the microcavity and molecular vibrational modes.

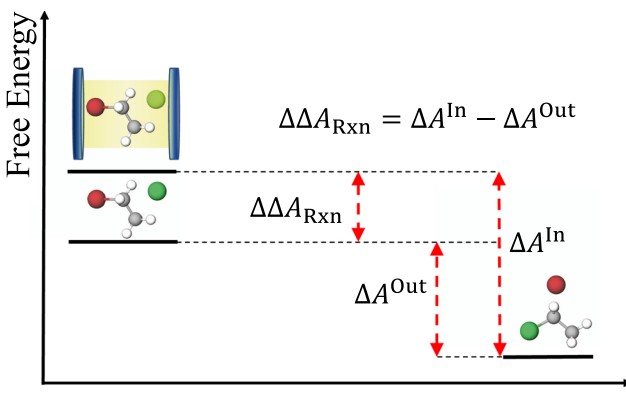

**Fig. 4 | Schematic representation of single-mode VSC effect on chemical equilibria.** The proposed theory indicates polariton-assisted chemical equilibrium modifications are induced by the change in reaction free energy due to the different light–matter interaction of reactants and products. Formally, for a system with fixed volume $V$ (for simplicity), $\Delta A^{\mathrm{Out}}$ is the reaction free energy in free space and $\Delta A^{\mathrm{In}}$ is the reaction free energy inside a microcavity, so $\Delta\Delta A_{\mathrm{Rxn}}$ is change in the reaction free energy induced by vibrational strong coupling. The figure illustrates the case where reactants are strongly coupled to the considered cavity mode. The reactant free energy is effectively raised leading to an increase in the reaction free energy, and a field-induced shift of the chemical equilibrium towards the products.

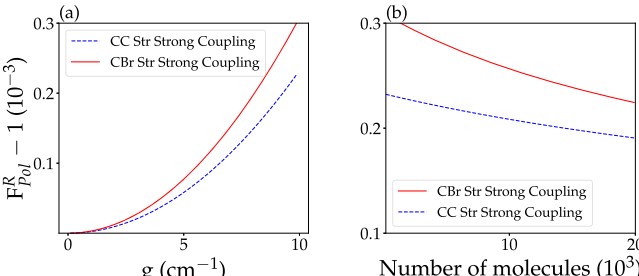

**Fig. 5 | System size and light–matter interaction strength dependence of single-mode vibrational strong coupling on a chemical equilibrium model. a** Polariton effect on reaction quotient (as measured by $F_{\mathrm{Pol}}^{\mathrm{R}} - 1$ in units of $10^{-3}$, see Eq. (40)) vs single-molecule light–matter coupling strength ($g$) for a system with a maximum number of 100 reactant molecules at $T = 300$ K. Each curve corresponds to vibrational strong coupling (VSC) between a particular molecular normal mode (CC Str in dashed blue and CBr Str in red) and a single cavity photon mode with the same frequency. **b** Polariton effect on equilibrium mixture composition (measured by $F_{\mathrm{Pol}}^{\mathrm{R}} - 1$ in units of $10^{-3}$, since here we restricted our attention to reactant VSC, see Eq. (40)) at $T = 300$ K and $g = 10$ cm$^{-1}$ vs total number of reactant and product molecules ($C_2H_5Br$ and $C_2H_5Cl$ respectively). Each curve corresponds to VSC between a particular reactant normal mode (CC Str in dashed blue and CBr Str in red) and a single cavity mode.

At higher temperatures, the polariton effect on the composition of the molecular mixture at thermodynamic equilibrium becomes negligible ($F_{\mathrm{Pol}} \to 1$) regardless of the vibrational frequency and light–matter coupling strength. The absence of any effect on the internal energy may be seen from the equipartition theorem (this implies that each normal mode has $k_B T$ mean internal energy)[1,2], whereas entropy variations induced by strong light–matter coupling may be seen to vanish from the classical limit of the harmonic oscillator partition functions which give

$$\lim_{T\to\infty} \Delta s \approx \frac{1}{N_F+1} \ln\left(\frac{q_{\mathrm{LP}} q_{\mathrm{UP}}}{q_F q_C}\right)$$
$$= \frac{1}{2(N_F+1)} \ln\left(\frac{\omega_C^2}{\omega_C^2 + \Omega_R^2 - \Omega_R^2}\right) = 0. \quad (34)$$

It follows that $T\Delta s$ goes to 0 at low and high $T$ but is an increasing function of $T$ at intermediate temperatures, therefore showing a maximum at moderate $T$ (Fig. 3).

Note the quantum treatment of field and molecular vibrational modes is essential, as the polariton-induced change in molecular free energy at the experimentally relevant temperatures $T \ll T_\nu$ is dictated by the Bose-Einstein distribution. Quantum statistics is relevant here, for the contribution of excited states to vibrational free energies is small for high-frequency normal modes with $T_\nu \gg T$, and exactly under such conditions, the Bose-Einstein distribution is significantly different from the classical Maxwell-Boltzmann. This explains the distinct polariton effects on thermodynamic properties of molecular systems observed here relative to those examined in the classical limit by Li et al.[35].

**Direction of chemical equilibrium shift.** Figure 3 also explains why single-mode strong coupling with a chemical species tends to bias the equilibrium towards the uncoupled species. This occurs because the sum of polariton zero-point energies $E_{\mathrm{LP}} + E_{\mathrm{UP}}$ is greater than the sum of the bare molecule normal-mode and bare photon zero-point energies. This feature increases the free energy of the light–matter system inside the microcavity relative to the bare system (Fig. 4).

**Magnitude of polariton effect on chemical equilibrium.** The single-mode strong coupling effect on chemical equilibrium as measured by the ratio of the reaction quotient in the microcavity to the standard-state (bare) equilibrium constant is observed to be less than 1.003 even at low temperatures such as 50 K. The effect becomes even weaker at higher temperatures, and may be understood from Fig. 3, which shows that the (single-cavity mode) polariton effect on the free energy per degree of freedom is tiny. We revisit this point when discussing the system-size dependence of our results later. Note the quantitative results present in this section do not rule out a polaritonic effect on chemical equilibria in complex EM environments where multiple vibrational modes of the reactive species strongly interact with a continuous set of on and off-resonance cavity modes as in a planar microcavity.

**Strong coupling with lower frequency normal modes have greater impact on equilibrium reaction quotients.** Figure 2 shows that strong light–matter coupling is most effective (among the scenarios we considered) when the matter part of polaritons corresponds to the CCl (product normal mode) or CBr (reactant normal mode) stretch modes. These motions have lower frequency than the CC stretch of either reactants or products by about 300 and 400 cm$^{-1}$, respectively. The greater impact of VSC occurring with lower frequency vibrations may be understood mathematically from an analysis of the polariton contribution to the zero-point energy difference between the polaritonic system and the composite (light–matter) bare system per (strongly coupled) degree of freedom. Under the conditions examined here where $\tilde{\omega}_C \approx \omega_M$ and $g/\omega_M \ll 1$, the polariton effect at the zero-point energy difference is given by

$$\lim_{T\to 0} \Delta e \approx \frac{g^2}{2\omega_M}, \quad g/\omega_M \ll 1. \quad (35)$$

This result clearly demonstrates that at low temperatures, where single-mode cavity effects on equilibrium are largest, light–matter interactions will have more significant impact when they involve vibrational modes with lower frequency and greater oscillator strength[36]. Similar results are valid at higher temperatures where thermal excitations play a greater role but ultimately lead to no polariton effect at chemical equilibrium in the $T \to \infty$ limit[36].

**Size and oscillator strength dependence of polariton effects on chemical equilibrium.** In the examined model reaction, all strongly coupled modes have nearly equal oscillator strength. However, this is not a generic feature of polyatomic molecules, which will generally have vibrational excitations with variable absorption intensity. In order to assess the dependence of $F_{Pol}$ on the single-molecule light–matter coupling strength, we present in Fig. 5(a) the behavior of $F_{Pol}$ at $T = 300$ K as a function of the single-molecule light–matter coupling constant. As expected (based on Eq. 35, see also Ref. 36), the polariton effect on the composition of the reactive mixture is enhanced with increasing single-molecule light–matter coupling strength.

We end our analysis of single-mode microcavity effects on chemical equilibrium by quantitatively investigating the behavior of the polariton effect under changes in the maximum number of strongly coupled molecules $N$ (with fixed cavity volume). We find that while $\Omega_R$ increases, the overall VSC effect on the reactive mixture composition decreases substantially as the number of molecules increases at $T = 300$K [Fig. 5(b)].

The weakening of VSC-induced changes on chemical equilibrium with increasing molecular density is an expected feature of single-cavity mode theories[35,72,73] which have systematically shown that polariton effects on local molecular observables decrease with increasing molecular density. Figure 5 shows a substantial deviation of the scaling with system size of the polariton effect on equilibrium reaction quotients relative to $1/N$. Still, the implication of various earlier studies[35,72,74–76] remains valid that at thermal equilibrium single-mode cavity effects on local molecular observables are insignificant in the collective light–matter interaction regime.

In conclusion, we provided a theory of chemical equilibrium under nonperturbative light–matter interactions. Using separability conditions motivated from the disparate timescales of slow and fast molecular degrees of freedom, we obtained a nonlinear relation between equilibrium reaction quotients inside and outside a microcavity (Eq. 25) based on the polariton effect on the chemical potential of each component of the reactive mixture.

We applied our theory to an $S_N2$ reaction in a single-mode cavity and found that polaritons can shift chemical equilibrium constants towards either direction of a reaction depending on the species (reactant or product) strongly coupled to the EM field, light–matter interaction effects are most impactful at lower temperatures, and the change induced by VSC on the internal energy of the light–matter system provide the dominant contribution to the VSC effect on chemical equilibria. We also showed that strong light–matter coupling is more effective at shifting chemical equilibria when polaritons are formed between IR cavity modes and molecular vibrations with significant oscillator strength and lower frequency. These trends were obtained in an idealized scenario where VSC mode occurs between a single EM mode of an IR resonator and particular normal modes of a particular component of the reactive mixture (reactant or product ensemble) but are based on fundamental features of our theory that are expected to hold more generally. Future work, based on Eq. (25) and discussed generalizations accounting for intermolecular interactions while including strong coupling of multiple IR modes of reactants and products with multimode Fabry-Perot cavities will allow direct quantitative comparison with experiments[3].

## Methods

### Numerical analysis of single-mode VSC effects on a model chemical equilibrium

When a single normal mode (of $N_F$ reactant or product molecules) interacts nonperturbatively with the optical microcavity, the polaritonic part of the Coulomb gauge[38] light–matter Hamiltonian can be written in the uncoupled basis as[7]

$$H = \sum_{i=1}^{N_F} \hbar\omega_F a_i^\dagger a_i + \hbar\tilde{\omega}_C b^\dagger b - ig\sqrt{\frac{\omega_F}{\tilde{\omega}_C}}\sum_{j=1}^{N_F}(a_j^\dagger - a_j)(b^\dagger + b), \quad (36)$$

where $\omega_F$ is the frequency of the strongly coupled molecular normal mode (of type F), $a_i^\dagger$ and $a_i$ are the creation and annihilation operators of F excitations in the $i$th molecule, and $b^\dagger$ and $b$ are the creation and annihilation operators of the cavity mode with renormalized frequency $\tilde{\omega}_C = \sqrt{\omega_C^2 + \Omega_R^2}$, where $\omega_C$ is the bare photon frequency and $\Omega_R = 2g\sqrt{N_F}$ is the collective light–matter interaction strength. We assume that the bare cavity mode is in resonance with a reactant or product normal mode (Table 1) and from now on set $\omega_C = \omega_F$. When $\tilde{\omega}_C$ is near-resonant with the molecular normal mode, the effective collective light–matter interaction strength for the strongly coupled species is $g\sqrt{N_F\omega_F/\tilde{\omega}_C} \approx g\sqrt{N_F}$. Note that the reactant and product modes in Table 1 have similar oscillator strength[71], and, therefore we employ the same value of $g$ when analyzing the effects on chemical equilibrium induced by exclusive strong light–matter coupling with each mode.

The light–matter system described by the Hamiltonian given by Eq. (36) has $N_F + 1$ eigenmodes. The frequencies of the hybrid excitations (polaritons) are

$$\omega_{LP} = \sqrt{\frac{\tilde{\omega}_C^2 + \omega_F^2 - \sqrt{\left(\tilde{\omega}_C^2 - \omega_F^2\right)^2 + 4\Omega_R^2\omega_F^2}}{2}}, \quad (37)$$

$$\omega_{UP} = \sqrt{\frac{\tilde{\omega}_C^2 + \omega_F^2 + \sqrt{\left(\tilde{\omega}_C^2 - \omega_F^2\right)^2 + 4\Omega_R^2\omega_F^2}}{2}}. \quad (38)$$

The remaining $N_F - 1$ normal modes form a degenerate purely molecular reservoir with the same frequency $\omega_F$ as the bare molecules.

Using basic statistical mechanics[1,2], we can obtain the polariton and reservoir mode partition functions and compute the polariton effect on the chemical potential $\Delta\mu_{F,Pol}$ (Eq. (22)) required to set up the nonlinear Eq. (25). Its solution consists of the equilibrium number of molecules of each species inside the optical cavity, and allows us to establish the polariton effect on the chemical equilibrium as measured by $F_{Pol}(V, T)$ via Eq. (27).

The numerical problem is set up by assuming that the mixture initially contains an equal number of ethyl bromide and chloride ions $N = N_{C_2H_5Br}^0 = N_{Cl^-}^0$ that react according to Eq. (29) to establish equilibrium with ethyl chloride and bromine ions. The number of reactant and product molecules at equilibrium is denoted $N_R$ and $N_P$ respectively. It follows that at equilibrium $N_R = N_{C_2H_5Br} = N_{Cl^-}$ and $N_P = N_{C_2H_5Cl} = N_{Br^-} = N - N_R$. The standard-state equilibrium constant $K_0(T)$ (outside the microcavity) is computed as a function of temperature using the ratio of product and reactant partition functions. To find the equilibrium composition of the mixture at thermal equilibrium, we solve the equation

$$\frac{(N - N_R)^2}{N_R^2} = K_0(T)e^{-\beta\tilde{\nu}_F[\mu_F^{LP}(N_F, V, T) + \mu_F^{UP}(N_F, V, T)]}, \quad (39)$$

for $N_R$, where $\tilde{\nu}_F$ is the signed stoichiometric coefficient of the strongly coupled species, and $N_F = N_R$ or $N_P$ when strong coupling occurs with a reactant or product normal mode, respectively. We solve Eq. (39) for a given $T$, initial number of molecules $N$, and single-molecule light–matter interaction strength $g$. The standard-state equilibrium composition of the reactive mixture is employed as an initial guess for the solution, and the polariton contributions to the chemical potential

**Article** https://doi.org/10.1038/s41467-024-46442-1

are obtained from automatic differentiation of the polariton free energies with respect to the number of strongly coupled molecules as implemented in the python AutoGrad package[77].

When a single reactant or product vibrational mode strongly interacts with the microcavity, the ratios of the reaction quotient under reactant and product strong coupling to the bare equilibrium constant are given respectively by

$$F_{\mathrm{Pol}}^{\mathrm{R}} = \exp\left[\beta\mu_{\mathrm{R,Pol}}(N_{\mathrm{R}})\right], \tag{40}$$

$$F_{\mathrm{Pol}}^{\mathrm{P}} = \exp\left[-\beta\mu_{\mathrm{P,Pol}}(N_{\mathrm{P}})\right], \tag{41}$$

where $\mu_{\mathrm{R,Pol}}(N_{\mathrm{R}}) = \mu_{\mathrm{R}}^{\mathrm{LP}}(N_{\mathrm{R}}) + \mu_{\mathrm{R}}^{\mathrm{UP}}(N_{\mathrm{R}})$ and a similar expression holds for the polaritonic contribution to the chemical potential of the product $P$.

## Molecular partition functions

Our results examine scenarios where $T$ is much smaller than the electronic excitation energies of all molecules involved. Therefore, only the electronic ground state of each chemical species is assumed to be occupied. Vibrational partition functions were constructed using the quantum harmonic oscillator model, whereas classical rotational and translational partition functions were employed for other degrees of freedom. Vibrational frequencies and moments of inertia were extracted from the Chemistry WebBook[71], while ground-state electronic energies were obtained from CCSD/aug-cc-pVTZ as given in the Computational Chemistry Comparison and Benchmark Database[78]. The following expressions for the translational, rotational, vibrational, and electronic partition functions of an asymmetric top molecule with $n$ normal modes were employed:

$$q_{\mathrm{trans}}(V, T) = \left(\frac{2\pi m\mathrm{k}_{\mathrm{B}}T}{h^2}\right)^{3/2} V, \tag{42}$$

$$q_{\mathrm{rot}}(T) = \frac{\pi^{1/2}}{\sigma}\prod_{j=1}^{3}\left(\frac{8\pi^2 I_j\mathrm{k}_{\mathrm{B}}T}{h^2}\right)^{1/2}, \tag{43}$$

$$q_{\mathrm{vib}}(T) = \prod_{a=1}^{n}\frac{e^{-\beta\hbar\omega_a/2}}{1 - e^{-\beta\hbar\omega_a}}, \quad q_{\mathrm{el}}(T) = e^{-\beta E_{\mathrm{g,el}}}, \tag{44}$$

where $m$ is the molecular mass, $V$ is the volume occupied by the system, $I_1$, $I_2$ and $I_3$ denote principal moments of inertia, $\sigma$ is the molecular symmetry number, $\omega_a$ is the $a$th normal-mode frequency, and $E_{\mathrm{g,el}}$ is the electronic ground-state energy.

## Reporting summary

Further information on research design is available in the Nature Portfolio Reporting Summary linked to this article.

## Data availability

The data generated in this study are provided in the Source Data file. Source data are provided with this paper.

## Code availability

The code used to generate all the numerical data presented here is available at https://github.com/RibeiroGroup/Chemical-Equilibria-in-Single-Mode-Cavity[79].

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

## Acknowledgements

R.F.R acknowledges financial support from the NSF CAREER Award (Grant No. CHE-2340746) and generous start-up funds from Emory University.

## Author contributions

R.F.R. coordinated this work, wrote the paper, performed the general mathematical derivations, analyzed the implications of these results, and contributed to the data analysis. K.S. helped to draft the paper, derived low and high-temperature limits, numerically implemented the developed formalism, prepared figures, and contributed to the analysis of the data.

## Competing interests

The authors declare no competing interests.
