## [Peer Review File · Nature Communications]

Reviewers' Comments:

Reviewer #1:

Remarks to the Author:

This paper presents a general study of chemical equilibrium under nonperturbative light-matter interactions. The authors derive an exact relationship between equilibrium reactive mixtures inside and outside of microcavities and illustrate their formalism for a SN2 reaction in a single-mode cavity. They find that polaritons can shift chemical equilibrium to both reactant or product when strongly coupled to the EM field (depending on the species). They also show that the effects are strongest at low temperature and the internal energy is mostly impacted by VSC. Overall, this is a very interesting and thoroughly executed new quantum theoretical approach, which is general and has the potential to shine some light on the many open questions regarding the understanding and the control of chemical equilibria with microcavities. This paper is suitable for the scope of Nature Communications and I am in favor of publishing this work after the authors addressed the following remarks:

- The authors discuss Eq.(19) and several of its important implications. Although I appreciate the detailed discussion, it is unclear which part of the equation implicates the mentioned features. If the authors could explicitly mention to which part/parameters they are referring to in Eq.(19), it would make the discussion much more accessible and easier to follow.
- The authors state that Eq.(19) highlights, that a multimode description of the electromagnetic field is essential for modeling the nonperturbative effect of planar microcavities on chemical equilibria. However, when they apply their equilibrium condition to the reactive mixture, they consider a coupling to only a single cavity-mode. Later they do refer to their Hamiltonian as the nonperturbative light-matter Hamiltonian. Can the authors comment on why a single mode treatment can still capture the nonperturbative effects as it seems to contradict their statement in the discussion of Eq.(19).
- As the authors mention, there has been other work on chemical equilibria in microcavities (e.g. they mention Li et al). However, most of them report small or no changes at all. Can the authors give some insight on why they can observe changes and (if possible to single out) what is the difference to work such as Li et al's?
- The authors explicitly call their approach a quantum theoretical approach, which to my understanding comes from the fact that they include the quantum mechanical description of the EM field in their theory (Eq.5). Do the authors believe that the quantum treatment of light is crucial in order to see the changes such as in Fig.3 ? If so I would recommend adding this to the discussion.
- In Fig.4 the authors show that the single mode cavity effect diminishes with increasing number of molecules. However, when considering the common microcavity experimental set ups, many molecules are needed to achieve strong coupling. Does that mean that the effects/changes the authors observe in this study will not hold (or are too small) in a realistic experimental set up?
- The authors mention that this approach is valid in generic electromagnetic environments, which is a very general statement. Can the authors elaborate what that exactly means? For example, is a description beyond the simple quantum harmonic oscillator possible, can longitudinal components of the EM field be included, is this method still computationally tractable for complex EM environments (multimode description), to name a few. A clarification of their statement could make this approach even more general.
- A minor point, but it is not straight forward to read the y-axis of Fig.3 and consequently to understand if these changes observed are actually large/small. Perhaps a little bit more detail in the caption (or label) might help to make it easier to read.

Reviewer #2:

Remarks to the Author:

The manuscript informs about calculations that connect reaction coefficients of different electromagnetic environments. The main result is that in the large N limit no effect on the reaction rate persists. This result reproduces the well established fact that the standard models of collective coupling, which the authors employ implicitly, are not able to capture the chemical effects.

Therefore I think that this work does not provide a significant step forward in describing or understanding the effects observed in experiments.

I am also unsure about the correctness of the obtained equations and thus results. The presentation makes it impossible for me to verify the steps of the authors. The starting equation (2) is largely undefined. The authors should write down their total Hamiltonian and then explain how they get to equation (6) step by step. Without stating explicitly the light-matter coupling Hamiltonian the whole derivation seems void. If I assume dipole coupling in the Power-Zienau-Woolley gauge, which is one of the many possible Hamiltonians of reference [38], then it is quite easy to see that at least rotations are definitely not decoupled from the light-matter coupling, as opposed to the statement of the authors. If all molecules are rotated perpendicular to the main cavity mode then there will be essentially zero coupling and thus the whole spectrum changes. That the rotations and with this the value of the coupling terms is important has been discussed by various authors {1,2,3}. What are the modes indicated by (k_x, k_y, k_z) and why can the modes in G_L be discarded? What is the meaning of "molecular infrared polarization operator is assumed to be linear"? Please provide an equation. Why should perfectly harmonic polariton modes emerge in any realistic ensemble of molecules? How does the number of molecules appear in equation (7)? Can equation (14) be derived instead of defined? The appearance of $N_A - 1$ dark modes is unambiguously true only for the Tavis-Cummings model. Why would that be a generic feature as seemingly claimed by the authors? Are all their derivations implicitly limited to the Tavis-Cummings model? The moment any disorder appears a strict distinction between dark and bright states becomes illegitimate. How do the specific partition functions that the authors use look like? There seems to be a typo in equation (23), because as is the Hamiltonian would not be hermitean. Why do the authors switch to a single-mode situation finally?

{1} Luk, Hoi Ling, et al. "Multiscale molecular dynamics simulations of polaritonic chemistry." *Journal of chemical theory and computation* 13.9 (2017): 4324-4335.

{2} Li, Tao E., Joseph E. Subotnik, and Abraham Nitzan. "Cavity molecular dynamics simulations of liquid water under vibrational ultrastrong coupling." *Proceedings of the National Academy of Sciences* 117.31 (2020): 18324-18331.

{3} Wang, Derek S., et al. "Cavity-modified unimolecular dissociation reactions via intramolecular vibrational energy redistribution." *The Journal of Physical Chemistry Letters* 13.15 (2022): 3317-3324.

Reviewer #3:

Remarks to the Author:

In this paper, the authors address the effects of light-matter coupling on the thermodynamics of a model reaction, namely the SN2 reaction of $C_2H_5Br + Cl^-$ to form $C_2H_5Cl + Br^-$. The authors focus on a cavity with a single high quality mode coupled to a single vibrational mode of either the reactants or the products. They explore the effects of the polaritons on the equilibrium constant under varied conditions of temperature, coupling mode (reactants or products), oscillator strength, and number of molecules. The main conclusions from this work include the polariton effects are strongest at low temperatures, equilibrium is shifted towards products when coupled to the reactants and towards reactants when coupled to products, the observed effects on the equilibrium are small in magnitude compared to the single molecule coupling strengths employed in the model, and polaritons formed between molecular modes with lower frequency have a stronger impact on the chemical equilibrium.

The authors accurately represent the work as contributing to an area of polariton research that has not been given as much attention in theoretical research and the authors cite the few other relevant sources that have contributed to this area. I believe this paper represents a novel and significant contribution to the field with sound methodology and conclusions that are well-supported and reproducible. I recommend publication with one minor comment. Figure 3 is a pivotal figure in explaining the results of this paper, but I think it is difficult to understand and draw the necessary conclusions from the figure alone. The text following the figure is helpful, but I think readers would benefit from additional graphics or text providing context for the trajectory of

these variables for this particular SN2 reaction without strong coupling. I would also suggest potentially breaking the figure down into two separate figures- one figure focusing on the change in these values at a particular temperature to highlight the influence of the polaritons relative to a reaction without strong coupling and then a separate figure to highlight the trajectory as a function of temperature.

Chemical equilibrium under vibrational strong coupling

Kaihong Sun and Raphael F. Ribeiro*

Department of Chemistry and Cherry Emerson Center for Scientific Computation, Emory University, Atlanta, GA, 30322

(Dated: September 13, 2023)

We introduce a theory of chemical equilibrium in optical microcavities, which allows us to relate equilibrium reaction quotients in different electromagnetic environments. Our theory shows that in planar microcavities under strong coupling with polyatomic molecules, hybrid modes formed between all dipole-active vibrations and cavity resonances contribute to polariton-assisted chemical equilibrium shifts. To illustrate key aspects of our formalism, we explore a model S_N2 reaction within a single-mode infrared resonator. Our findings reveal that chemical equilibria can be shifted in either direction of a chemical reaction, depending on the oscillator strength and frequencies of reactant and product normal modes. Polariton-induced zero-point energy changes provide the dominant contributions, though the effects in idealized single-mode cavities tend to diminish quickly as the temperature and number of molecules increase. Our approach is valid in generic electromagnetic environments and paves the way for understanding and controlling chemical equilibria with microcavities.

I. INTRODUCTION

Light-matter interactions are almost always irrelevant in equilibrium thermodynamics [1, 2]. However, recent experiments have suggested otherwise, that the chemical equilibrium of aromatic-halogen charge-transfer complexes may be significantly changed via strong light-matter coupling [3].

The signature of strong light-matter interactions is the formation of hybrid states referred to as polaritons, consisting of a superposition of electromagnetic (EM) and matter excitations [4–6]. Devices that confine the EM field to the scale of relevant wavelengths [e.g., for infrared (IR) strong coupling, planar cavities are generally constructed with moderate quality mirrors separated by a distance of $O(\mu\text{m})$] [7–11] are generally conducive to polariton formation in the presence of a resonant material (Fig. 1). A simple paradigmatic model of this phenomenon includes an isolated cavity mode under strong interaction with the collective polarization of a molecular system containing N identical molecules. This system has two hybrid light-matter modes denoted lower and upper polaritons (LP and UP, respectively), and $N - 1$ molecular reservoir modes with zero photonic content.

Recent experimental reports have provided evidence that chemical reactions can be substantially affected by strong interactions between IR microcavities and near-resonant molecular vibrational normal modes (vibrational strong coupling) [12–19]. Modulation of charge conductivity [20–22], and energy transport phenomena [23–27] have also been reported.

While theoretical investigations have proposed hypothetical mechanisms for microcavity effects on reaction rates via nonequilibrium effects [28–33], less attention has been paid to polariton effects on thermodynamic quantities of molecular systems. Scholes et al. [34] showed the

FIG. 1: Schematic representation of a reactive mixture in an infrared microcavity supporting confined electromagnetic field modes and strong light-matter interactions, which lead to the formation of hybrid polariton normal-modes with distinct spectra relative to the molecular system in free space and the empty microcavity.

free energy of dark modes is lower than the polaritonic, and Li et al. [35] employed classical statistical mechanics to argue that collective strong light-matter coupling is unlikely to affect molecular potentials of mean force. However, recent quantum approaches have shown that polariton effects on thermodynamic quantities could be significant, especially under ultrastrong coupling conditions [36, 37].

In this work, we present a quantum theoretical investigation of chemical equilibrium under vibrational strong coupling (VSC). We provide a general theory of non-perturbative light-matter interaction effects on chemical equilibrium and obtain an exact relationship between equilibrium reactive mixtures inside and outside microcavities in Sec. II. In Sec. III, we apply our theory to a multicomponent reactive mixture in a single-mode IR cavity resonant with a bright normal-mode of reactants or products. We examine the temperature, normal mode frequency, oscillator strength, and size dependence of the polariton effect on the reactive mixture composition at equilibrium. We summarize our main results and explain how the provided formalism informs future work in Sec. IV.

* raphael.ribeiro@emory.edu

II. THEORY

In this section, we present a general formalism for the investigation of nonperturbative light-matter interaction effects on the composition of reactive mixtures. Let A, B, C, and E denote reactive chemical species in equilibrium (in the gas-phase for simplicity) according to

The total molecular quantum electrodynamic Hamiltonian [38] for this system in the Coulomb gauge is given by

$$H = H_M + H_L + H_{LM}, \quad (2)$$

where H_L is the transverse EM field Hamiltonian that generates the free field dynamics in terms of the mode creation and annihilation operators $a_{\mathbf{k}\lambda}^\dagger$ and $a_{\mathbf{k}\lambda}$, respectively

$$H_L = \sum_{\mathbf{k}\lambda}^{k < k_M} \hbar \omega_{\mathbf{k}} \left(a_{\mathbf{k}\lambda}^\dagger a_{\mathbf{k}\lambda} + \frac{1}{2} \right) \quad (3)$$

where $\mathbf{k} = (k_x, k_y, k_z)$ is the wave vector with components k_x , k_y , and k_z fulfilling boundary conditions associated with the electromagnetic environment [39], $k = |\mathbf{k}|$, $\lambda = 1, 2$ denotes the field polarization, and k_M is a high-energy cutoff for photon modes. Specifically, in the treatment detailed below, only photon modes with $k < k_M$ are assumed to form polaritons. In any particular application k_M would depend on the molecular system considered and the strength of the collective light-matter interactions [40].

The pure matter part of the Hamiltonian is denoted by H_M and given by

$$H_M = h_M + V_M, \quad (4)$$

where V_M corresponds to the (longitudinal) intermolecular electrostatic interactions (between electronic and nuclear charges of different molecules), and h_M is the Hamiltonian for a noninteracting mixture of A,B,C and E molecules

$$h_M = h_A + h_B + h_C + h_E. \quad (5)$$

The noninteracting subsystem Hamiltonian h_F corresponds to the nonrelativistic electrostatic Hamiltonian describing a pure ensemble of N_F noninteracting molecules of type $F \in \{A, B, C, E\}$, i.e., $h_F = \sum_{i=1}^{N_F} h_{iF}$, where

$$h_{iF} = \sum_{\alpha} \frac{\mathbf{p}_{i\alpha}^2}{2m_{i\alpha}} + V_{iF}^{\text{Coul}}, \quad (6)$$

$\mathbf{p}_{i\alpha}$ is the kinetic momentum of the α charge of the i th molecule, $m_{i\alpha}$ is the corresponding rest mass, and V_{iF}^{Coul}

is the intramolecular longitudinal electrostatic interactions of electrons and nuclei of molecule i in the noninteracting subensemble of type F . From now on, we consider all molecules involved are nonlinear and polyatomic for the sake of simplicity. Restrictions to the cases where any of the reactants or products are monoatomic, diatomic or linear molecules can be straightforwardly performed as needed for any application based on the treatment below.

We approximate the light-matter interaction Hamiltonian H_{LM} in the Coulomb gauge as follows:

$$H_{LM} = \sum_{\zeta} [\mathbf{j}_{\zeta} \cdot \mathbf{A}(\mathbf{X}_{\zeta}) + \tilde{e}_{\zeta}^2 \mathbf{A}^2(\mathbf{X}_{\zeta})], \quad (7)$$

where each molecule is denoted by ζ , \mathbf{X}_{ζ} is its center of charge, e_{α} is the charge of particle α (nucleus or electron), $\mathbf{j}_{\zeta} = \sum_{\alpha} e_{\alpha} \mathbf{p}_{\alpha} / m_{\alpha}$ is the charge current operator in the lab (space) frame (with fixed axes) and $\tilde{e}_{\zeta}^2 = \sum_{\alpha} e_{\alpha}^2 / (2m_{\alpha})$. This Hamiltonian employs a long-wavelength limit approximation that neglects inhomogeneities of the EM field on the spatial scale of the molecular charge density. This poses no concern because the infrared field modes of relevance to us have wavelength that is orders of magnitude larger than typical molecular sizes.

Our description to this point has been extremely general, but in what follows we specialize to the case where nonperturbative light-matter interactions occur exclusively between high-frequency vibrational normal-modes of the molecular ensemble and the confined electromagnetic modes here considered. We will only be concerned with VSC effects on chemical equilibria at temperatures T that are (a) much greater than the rotational temperatures T_{rot} of all involved molecules, (b) much lower than the lowest lying electronic transition of each considered chemical species (at their equilibrium geometry), and (c) much lower than the vibrational temperatures $T_{\nu} = h\nu/k_B$ of the bright (dipole-active) normal-modes of each molecule (here labeled by the frequency ν). These conditions are generally satisfied by polyatomic molecules at standard conditions of temperature and pressure [1, 2], including those employed in VSC experiments that reported polariton generation by interaction of infrared cavity modes with high-frequency molecular vibrations of polyatomic molecules satisfying $T_{\nu} \gg T \gg T_{\text{rot}}$ [41–45].

Several important implications follow from the conditions (a), (b) and (c) given above. From (a) ($T \gg T_{\text{rot}}$), it follows that a classical statistical mechanical treatment of the molecular rotational degrees of freedom provides a reliable approximation to the rotational contribution to the thermodynamic observables of the considered system [1, 2]. Therefore, hereafter we consider the rotational degrees of freedom of h_M to be classical phase space variables [46]. Similar considerations can be made for the molecular translational degrees of freedom, for we only examine systems where intermolecular distances are much larger than the thermal de Broglie wavelength of

each chemical species. This allows us to treat molecular translational degrees of freedom as classical variables. Note this partitioning of molecular degrees of freedom into classical and quantum does not pose any practical, nor conceptual challenge, especially as rotational and translational motions occur on timescales that are orders of magnitude slower than the vibrational dynamics involved in VSC. Further, our treatment is such that in the limit where the light-matter coupling goes to zero, we recover the standard predictions of the statistical mechanics of polyatomic systems, where vibrational modes are treated quantum mechanically and rotations, translations, librations, etc are treated classically [1, 2].

Condition (b) ($k_B T$ is much less than the lowest-lying electronic transition energy of each molecule), implies then that the relevant eigenstates of the noninteracting molecular Hamiltonian h_{iF} are products of the F electronic ground-state and vibrational wave functions. Further simplification arises from condition (c) that $T \ll T_\nu$ for the molecular normal-modes under strong interaction with the EM field. In particular, this condition guarantees negligible thermal populations for states of dipole-active vibrational modes with more than a single quantum (only ground and first excited-states are occupied with any significance) at experimentally probed temperatures, so the anharmonicity of modes under VSC is inconsequential for thermal properties of the molecular system, and we can accurately treat the molecular high-frequency modes under VSC as harmonic oscillators without loss of any significant accuracy. Notably, this same argument allows one to treat anharmonic high-frequency modes of molecules as harmonic oscillators and still obtain great agreement with experiments [1].

The fact that vibrational modes with significant oscillator strength satisfy $T_\nu \gg T$ allows significant simplification of their thermodynamic treatment. In particular, the assumed absence of large-amplitude anharmonic motion implies that the approximate separation of vibrational and rotational motions in the Eckart frame [47–49] is reliable and independent normal-coordinates may be assigned to high-frequency vibrational modes executing small-amplitude motions determined by the lowest vibrational states (for $T \ll T_\nu$, only the ground and first-excited state associated to each bright normal-mode have significant thermal occupation [1, 2]). In this case, the molecular infrared polarization operator $\mathbf{P}(\mathbf{x}) = \sum_\zeta \mathbf{P}_\zeta(\mathbf{x})$ can be accurately approximated by retaining only the constant and linear terms in its expansion in powers of the relevant normal-mode displacements (those associated to transitions with significant oscillator strength) $(\mathbf{Y}_{1\zeta}, \mathbf{Y}_{2\zeta}, \dots, \mathbf{Y}_{n_\zeta, \zeta})$ [50], where each of the normal displacements is written in the molecular frame (i.e., that rotates with the molecular system), n_ζ is the number of ζ normal-modes with significant oscillator strength ($n_\zeta \leq 3N_{a\zeta} - 6$ for each molecule ζ with $3N_{a\zeta}$ atoms and $3N_{a\zeta} - 6$ normal-modes in its electronic ground-state, as we have for simplicity assumed all involved molecules are nonlinear). Using the electrical dipole approximation to

the molecular polarization operator [38], it follows that $\mathbf{P}_\zeta(\mathbf{x})$ can be written as

$$\begin{aligned} \mathbf{P}_\zeta(\mathbf{x}) &= R_\zeta^T(\boldsymbol{\theta}_\zeta) [\mathbf{d}_{0\zeta} + \mathbf{d}_\zeta(\mathbf{Y}_\zeta)] \delta(\mathbf{x} - \mathbf{X}_\zeta), \quad (8) \\ \mathbf{d}_\zeta(\mathbf{Y}_\zeta) &= \sum_{i=1}^{n_\zeta} (\mathbf{Y}_{i\zeta} \cdot \nabla_Y) \mathbf{p}_\zeta(\mathbf{Y}) \Big|_{\mathbf{Y}=0}, \quad (9) \end{aligned}$$

where $\boldsymbol{\theta}_\zeta$ denotes the Euler angles specifying the orientation of molecule ζ , $R_\zeta^T(\boldsymbol{\theta}_\zeta)$ is the SO(3) rotation that maps the ζ frame into the lab frame [46], $\mathbf{d}_{0\zeta}$ is the molecular dipole moment operator at its equilibrium geometry ($\mathbf{Y} = 0$) in the molecular frame, and $\mathbf{Y}_{i\zeta} \cdot \nabla_Y$ is the nuclear gradient along the normal-mode displacement $\mathbf{Y}_{i\zeta}$ in the molecular frame.

The transition-matrix elements associated to the molecular current operators \mathbf{j}_ζ can be related to the matter polarization contribution from each molecule using the identity $\mathbf{j}_\zeta = \sum_\alpha e_\alpha \mathbf{p}_{\zeta\alpha} / m_{\zeta\alpha} = -i \sum_\alpha e_\alpha [x_\alpha, h_\zeta] / \hbar$. In the basis of vibrational eigenstates of h_ζ with corresponding eigenvalues $E_{\zeta a}$, Eqs. 8 and 9 imply $\mathbf{j}_\zeta^{ab} = R_\zeta^T(\boldsymbol{\theta}) \left[i\omega_\zeta^{ab} \mathbf{d}_\zeta^{ab}(\mathbf{Y}) \right]$, where $\omega_\zeta^{ab} = (E_{\zeta a} - E_{\zeta b}) / \hbar$. The contribution of each molecule to the diamagnetic contribution to H_{LM} (Eq. 7), namely $\bar{e}_\zeta^2 \mathbf{A}^2(\mathbf{X}_\zeta)$ must also be reconsidered in light of the restriction of our nonperturbative treatment to infrared molecular transitions, e.g., the Thomas-Reiche-Kuhn sum rule [51–53] can be employed to obtain in the restricted molecular Hilbert space a diamagnetic term that is consistent with the approximations here employed to model VSC.

The described framework, where high-frequency vibrations and EM modes are treated quantum mechanically, electronic degrees of freedom are frozen in the ground-state, and translations and rotations are treated classically leads to a family of quadratic Hamiltonians $h = h_M(\mathbf{J}, \mathbf{P}_C) + H_L + H_{LM}(\boldsymbol{\theta}, \mathbf{X})$ parametrized by the center of mass position and momentum of each molecule $(\mathbf{X}_\zeta, \mathbf{P}_{C\zeta})$ and their corresponding (classical) orientation and angular momentum $(\boldsymbol{\theta}_\zeta, \mathbf{J}_\zeta)$. Each set of classical molecular variables leads to a unique Hamiltonian h with two classes of eigenmodes: (i) polaritons with frequency $\omega_\alpha(\boldsymbol{\theta})$ (these could include modes with very negligible fraction of EM or molecular excitation) and (ii) dipole-inactive vibrations described by the Hamiltonian H_D which also includes the ground-state electronic energy and classical translational and rotational kinetic energy of each molecule. In the basis of eigenstates of $h_M(\mathbf{J}, \mathbf{P}_C) + H_L + H_{LM}(\boldsymbol{\theta}, \mathbf{X})$, the total Hamiltonian H (Eq. 2) can be written as

$$H(\mathbf{J}, \boldsymbol{\theta}, \mathbf{P}_C, \mathbf{X}) = H_{\text{Pol}}(\boldsymbol{\theta}, \mathbf{X}) + H_D(\mathbf{J}, \mathbf{P}_C) + V_M(\boldsymbol{\theta}, \mathbf{X}), \quad (10)$$

where $H_{\text{Pol}}(\boldsymbol{\theta})$ represents the (in general, multimode) vibrational polariton Hamiltonian in the normal-mode representation, H_D is the sum of (classical) molecular translational and rotational kinetic energies and dipole-inactive (normal-mode quantum) vibrational Hamiltonians, and $V_M(\boldsymbol{\theta}, \mathbf{X}_\zeta)$ describes intermolecular interactions.

The polaritonic part of the Hamiltonian can be expressed in terms of polariton creation and annihilation operators as follows:

$$H_{\text{Pol}} = \sum_{\alpha} \hbar\omega_{\alpha} \left(c_{\alpha}^{\dagger} c_{\alpha} + \frac{1}{2} \right), \quad (11)$$

where c_{α}^{\dagger} (c_{α}) is the bosonic creation (annihilation) operator associated with polariton mode α . Note that H_{Pol} , c_{α}^{\dagger} , c_{α} , and ω_{α} are all dependent on the molecular orientations θ_{ζ} , center of mass positions \mathbf{X}_{ζ} and number of molecules of each chemical species contributing to the formation of the polariton mode here labeled as α .

Intermolecular interactions represented by V_M include nonlinear couplings between degrees of freedom with free dynamics generated by both H_D and H_{Pol} . These interactions are known to contribute to polaritonic linewidths and to a reduction their lifetimes [54, 55]. For the sake of simplicity, we will proceed with the assumption that the intermolecular interactions modeled by V_M can be ignored for the purpose of computing the thermodynamic equilibrium properties of the molecular system. This assumption can and will be lifted later. Therefore, we employ

$$h(\mathbf{J}, \boldsymbol{\theta}, \mathbf{X}, \mathbf{P}_C) \equiv H_{\text{Pol}}(\boldsymbol{\theta}, \mathbf{X}) + H_D(\mathbf{J}, \mathbf{P}_C) \quad (12)$$

to compute the partition function of the light-matter system and its corresponding thermal properties.

The partition function of the light-matter system at fixed volume V , temperature T , and N_A, N_B, N_C, N_E molecules can be written as

$$Q(N, V, T) = \int \frac{d\mathbf{J}d\boldsymbol{\theta}d\mathbf{X}d\mathbf{P}_C}{(2\pi\hbar)^{12N_T}} \text{Tr} \left[e^{-\beta h(\mathbf{J}, \boldsymbol{\theta}, \mathbf{X}, \mathbf{P}_C)} \right] \quad (13)$$

where $N = (N_A, N_B, N_C, N_E)$, $d\mathbf{J}d\boldsymbol{\theta}$ is a compact notation for the $12N$ -symplectic form associated to the phase space translational and rotational variables of all molecules, $N_T = N_A + N_B + N_C + N_E$, and Tr denotes the trace over quantum states of all intramolecular vibrational degrees of freedom. The integration over \mathbf{J} and \mathbf{P}_C is trivial, for the Hamiltonian is quadratic in these variables. Further, we make the approximation that the molecular orientational and positional distributions are isotropic and uniform, respectively and unaffected by the interaction with the considered EM modes. These considerations imply that under the collective light-matter interaction regime, the molecular rotational and translational partition functions are unaffected by VSC, and the total partition function of the light-matter system can be approximated as

$$Q(N, V, T) = Q_{\text{Pol}}(N, V, T)Q_D(N, V, T), \quad (14)$$

where Q_D denotes the partition function for the electronic-translational-rotational and non-dipole active vibrational degrees of freedom

$$Q_D(N, V, T) = \prod_{\text{F}} \frac{[q_{\text{el}}^{\text{F}}(T)q_{\text{trans}}^{\text{F}}(V, T)q_{\text{rot}}^{\text{F}}(T)\tilde{q}_{\text{vib}}^{\text{F}}(T)]^{N_{\text{F}}}}{N_{\text{F}}!}, \quad (15)$$

where $\tilde{q}_{\text{vib}}^{\text{F}}(T)$ is the partition function associated with the dipole-inactive normal modes of a single F species.

The polaritonic partition function Q_{Pol} is obtained from the average over the orientation and center of mass coordinates of each molecule, i.e., $(2\pi\hbar)^{6N_T}Q_{\text{Pol}} = \int d\boldsymbol{\theta}d\mathbf{X} \text{Tr} \exp[-\beta H_{\text{Pol}}(\boldsymbol{\theta}, \mathbf{X})]$. In the collective light-matter interaction regime, polariton energies ω_{α} depend on the orientation and center of mass coordinates of a large ensemble of molecules. Thus, collective strong coupling suggests the fluctuations of the spectrum of $H_{\text{Pol}}(\boldsymbol{\theta}, \mathbf{X})$ from its configurational average are negligible [56, 57], and this implies a mean-field approximation to the light-matter Hamiltonian gives a reliable estimate of Q_{Pol} . In particular, Q_{Pol} can be approximated as $\text{Tr} \exp[-\beta \langle H_{\text{Pol}}(\boldsymbol{\theta}, \mathbf{X}) \rangle_{\boldsymbol{\theta}, \mathbf{X}}]$, where $\langle H_{\text{Pol}}(\boldsymbol{\theta}, \mathbf{X}) \rangle_{\boldsymbol{\theta}, \mathbf{X}}$ is the translational-orientational average of the polariton Hamiltonian [56]. This leads to Q_{Pol} corresponding to a product of polaritonic partition functions $\bar{q}_{\alpha}(T)$ with frequencies $\bar{\omega}_{\alpha}$ originating from averaging $H_{\text{Pol}}(\boldsymbol{\theta}, \mathbf{X})$ over isotropic distribution of molecular orientations and uniform distribution of the molecular center of mass coordinates. Note the mean-field frequencies $\bar{\omega}_{\alpha}$ are exactly those generally observed in linear optical response measurements of VSC in liquid or amorphous phases, where the probed molecular ensembles are in general isotropic, homogeneously distributed (if these properties also hold for the same molecular system in free space) and with spectrum showing negligible fluctuations due to the macroscopic nature of the ensemble.

The Helmholtz free energy of the light-matter system can now be directly obtained from $Q(N, V, T)$ as

$$\begin{aligned} A(N, V, T) &= -k_{\text{B}}T \ln[Q(N, V, T)], \\ &= A_{\text{Pol}}(N, V, T) + A_D(N, V, T), \end{aligned} \quad (16)$$

where $A_{\text{Pol}}(N, V, T) = -k_{\text{B}}T \ln[Q_{\text{Pol}}(N, V, T)]$ with $Q_{\text{Pol}}(N, V, T) = \text{Tr}[\exp(-\beta H_{\text{Pol}})]$ and corresponding expression applies for $A_D(N, V, T)$.

The chemical equilibrium condition at fixed V and T is [1, 2]

$$\begin{aligned} \sum_{\text{F}} \nu_{\text{F}} \frac{\partial A}{\partial N_{\text{F}}} &= -\nu_{\text{A}}\mu_{\text{A}} - \nu_{\text{B}}\mu_{\text{B}} + \nu_{\text{C}}\mu_{\text{C}} + \nu_{\text{E}}\mu_{\text{E}} \\ &= 0, \end{aligned} \quad (17)$$

where the chemical potential of each molecular species $\mu_{\text{F}}(N, V, T) = \partial A / \partial N_{\text{F}}$ is given by

$$\mu_{\text{F}}(N, V, T) = \mu_{\text{F,Pol}}(N, V, T) + \mu_{\text{F,D}}(N_{\text{F}}, V, T), \quad (18)$$

with $\mu_{\text{F,Pol}}(N, V, T) = \partial A_{\text{Pol}} / \partial N_{\text{F}}$ and $\mu_{\text{F,D}}(N_{\text{F}}, V, T) = \partial A_D / \partial N_{\text{F}}$. We can rewrite Eq. 18 in terms of a bare contribution and a polariton-induced change by adding and subtracting the contribution to the chemical potential from the bright vibrational part of h_{F} which we write as $\mu_{\text{F,vib, bright}}(T)$. Noting that $\mu_{\text{F,vib, bright}}(T)$ is the contribution of bright vibrational modes to the bare chemical potential for a system of noninteracting N_{F} molecules of

type F, and the remaining additive contribution to the F chemical potential is $\mu_{F,D}$, we define the reference chemical potential of species F by

$$\mu_F^{(0)}(N_F, V, T) = \mu_{F,D}(N_F, V, T) + \mu_{F,\text{vib,bright}}(T). \quad (19)$$

Note that $\mu_F^{(0)}(N_F, V, T)$ is employed as a standard-state relative to which the molecular chemical potential is obtained under conditions where interaction of the molecular system with the electromagnetic field may be significant. Equivalently, $\mu_F^{(0)}$ follows from the same separability conditions employed to obtain Eq. 14 in the limit where the light-matter interaction approaches zero. It follows that the chemical potential of species F under the influence of the EM field is given by

$$\mu_F(N, V, T) = \mu_F^{(0)}(N_F, V, T) + \Delta\mu_{F,\text{Pol}}(N, V, T), \quad (20)$$

where we introduced

$$\Delta\mu_{F,\text{Pol}} = \mu_{F,\text{Pol}} - \mu_{F,\text{vib,bright}}. \quad (21)$$

The bare F chemical potential can be obtained directly from Eqs. 19 and 15 as

$$\begin{aligned} \mu_F^{(0)}(N_F, V, T) &= -k_B T \frac{\partial}{\partial N_F} \ln \left[\frac{q_F^{N_F}(V, T)}{N_F!} \right] \\ &= -k_B T \ln \left[\frac{q_F(V, T)}{N_F} \right], \quad N_F \gg 1. \end{aligned} \quad (22)$$

where $q_F(V, T)$ is the single-molecule partition function for bare isolated species F and to obtain the second line we employed Stirling's approximation. By applying Eqs. 22 and 20 into Eq. 17, we obtain

$$-k_B T \sum_F \nu_F \ln \left[\frac{q_F}{N_F} \right] + \sum_F \nu_F \Delta\mu_{F,\text{Pol}} = 0. \quad (23)$$

A simple rearrangement leads to our expression for the equilibrium reaction quotient (Eq. 24) under the influence of nonperturbative light-matter interactions

$$\frac{N_E^{\nu_E} N_C^{\nu_C}}{N_A^{\nu_A} N_B^{\nu_B}} = \frac{q_E^{\nu_E} q_C^{\nu_C}}{q_A^{\nu_A} q_B^{\nu_B}} \exp \left[-\beta \sum_F \nu_F \Delta\mu_{F,\text{Pol}}(N, V, T) \right]. \quad (24)$$

By solving Eq. 24 for the number of molecules of each species under constraints set by the experimental situation (e.g., the system is initially prepared with an equal number of A and B molecules, etc), we obtain the polariton effect on the equilibrium composition of the reactive mixture.

Discussion. Generic properties of vibrational polariton effects on chemical equilibria arising as a consequence of Eq. 24 are summarized here. First, Eq. 24 shows a proper description of polariton effects on chemical equilibria requires a multimode description of the EM field, as both on and off-resonant modes contribute to the

VSC-induced changes in the reaction quotient. Explicitly, in a system with N_P eigenmodes (Eq. 11), we can take advantage of the assumed condition that $T_\nu \gg T$ (for modes strongly coupled to light) and the consequent quadratic nature of the strong light-matter system to obtain $Q_{\text{Pol}} = \prod_{l=1}^{N_P} q_{P_l}$, where q_{P_l} is the harmonic oscillator partition function associated to the l th polariton mode. It follows the polariton contribution to the free energy is additive, with $A_{\text{Pol}} = \sum_{l=1}^{N_P} A_{P_l}$, where $A_{P_l} = -k_B T \ln(q_{P_l})$. Hence, the polariton change in chemical potential $\Delta\mu_{F,\text{Pol}}$ has contributions from all N_P polariton modes with participation of chemical species F . In fact, using $\Delta\mu_{F,\text{Pol}} = \sum_{l=1}^{N_P} \mu_{F,P_l} - \mu_{F,\text{vib,bright}}$, where $\mu_{F,P_l} = \partial_{N_F} A_{P_l}$, we find directly that the r.h.s of Eq. 24 depends on *all* N_P polariton modes formed via hybridization with any of the molecular species. Clearly, no a priori special role is played by field fluctuations corresponding to incidence angles near zero, and devices with greater density of polariton modes will allow greater control of chemical equilibria. Additionally, it is seen that polaritons originating from all bands of a microcavity in resonance or sufficiently close to resonance with dipole-active molecular vibrations will contribute to Eq. 24.

Second, Eq. 24 demonstrates that in a polyatomic system with multiple bright vibrations, the chemical equilibrium shift induced by an IR microcavity depends on the density of EM modes at the various bright IR resonances of *both* reactants and products and their corresponding oscillator strengths. For instance, if the chemical species F has n_F bright normal-modes in resonance or near-resonance with EM modes corresponding to any incidence angle, then $\mu_{F,\text{vib,bright}}$ has additive contributions from all bright normal-modes $m = 1, 2, \dots, n_F$, i.e., $\mu_{F,\text{vib,bright}} = \sum_m \mu_F^{(m)}$, where $\mu_F^{(m)}$ is the chemical potential associated to the m th vibrational mode of each molecule, and $\Delta\mu_{F,\text{Pol}}$ will be impacted by polaritons formed between the F chemical species and the confined EM field. Again, here, there is, a priori, no special role played by the characteristics of the electromagnetic modes (e.g. the photon in-plane momentum in the case of a planar microcavity).

Eq. 24 also indicates that VSC may shift the equilibrium towards products or reactants depending only on their oscillator strength density, vibrational resonance frequencies, and the spectrum of the confined EM field. These quantities control the polariton contributions to the chemical potential $\Delta\mu_{F,\text{Pol}}$ via its dependence on the polariton energies, as these are determined by the collective interaction strengths of the various bright vibrational modes involved in a typical equilibrium. In fact, the VSC effect on equilibrium will lead to a greater fraction of product species (in comparison to free space under the same conditions), when the polariton-induced variation of the chemical potential of the products $\Delta\mu_{P,\text{Pol}} = \nu_C \Delta\mu_{C,\text{Pol}} + \nu_D \Delta\mu_{D,\text{Pol}}$ is *less* than the corresponding quantity for the reactants $\Delta\mu_{R,\text{Pol}} = \nu_A \Delta\mu_{A,\text{Pol}} + \nu_B \Delta\mu_{B,\text{Pol}}$, for in this case $\exp[-\beta(\Delta\mu_{P,\text{Pol}} - \Delta\mu_{R,\text{Pol}})]$ is greater than one, so it

follows from Eq. 24 that the reaction quotient under VSC as expressed by $N_C^{\nu_C} N_E^{\nu_E} / [N_A^{\nu_A} N_B^{\nu_B}]$ is greater than the reaction quotient in free space $[q_C^{\nu_C} q_E^{\nu_E} / (q_A^{\nu_A} q_B^{\nu_B})]$. In the next section, we quantitatively investigate $\Delta\mu_{F,Pol}$ in an elementary model of strong light-matter coupling to confirm the validity of these statements.

Our results have relied primarily on conditions accessed by the vast majority of VSC experiments (moderate temperatures that are much smaller than all electronic energy transition as well as the vibrational temperatures of high-frequency modes contributing to polaritons, but also much greater than rotational temperatures and isotropic molecular distribution of molecular ensembles negligibly perturbed by interaction with both polarizations of the EM field). We ignored the intermolecular term V_M and anharmonicity even in low-frequency vibrational modes (assumed to be weakly coupled to the confined field) to obtain Eq. 24, but these approximations can be easily made much less extreme without almost any change in our formalism. For instance, we can add intramolecular anharmonicity to the low-frequency modes without any change to Eq. 24 by employing an anharmonic vibrational partition for $\tilde{q}_{vib}(T)$. This could include nonlinear couplings between modes that are not involved in polariton formation, or anharmonic interactions that only significantly perturb highly-excited polariton modes (with at least $v \geq 2$) with negligible thermal occupation at the considered temperatures $T \ll T_v$. Likewise, we can reintroduce without any additional complexity, the effects of the intermolecular longitudinal electrostatic interactions V_M on the rotational, translational and vibrational modes with weak or vanishing oscillator strength. This procedure would lead to a new standard-state for the reaction quotient outside a microcavity, i.e., $q_E^{\nu_E} q_C^{\nu_C} / (q_A^{\nu_A} q_B^{\nu_B})$ would be converted into the expression of the equilibrium reaction quotient in free space accounting for the considered longitudinal interactions between all present chemical species.

Changes in the longitudinal EM interactions induced by the confined EM field [39] could be accounted for by writing $V_M = V_M^0 + \Delta V_M$, where V_M^0 is the free-space electrostatic potential and ΔV_M accounts for the renormalization of the free-space Coulomb potential induced by the boundary conditions satisfied by the EM field [37]. Explicit inclusion of this term would lead to another contribution to the field-matter change in the chemical potential of each species in the reactive mixture. Note that our main result makes no simplification nor assumption about the existence of energetic disorder which may weakly perturb normal-mode frequencies and change the equilibrium reaction quotient (Eq. 24) via the disorder-induced variation of $\Delta\mu_{F,Pol}$. In the simplest case where molecular interactions with an inert background lead to static disorder corresponding to small fluctuations in normal-mode frequencies, renormalized thermal observables could be obtained from the partition function resulting from the disorder-average of the light-matter $Q(N, V, T; \xi)$ obtained at a particular disorder

realization ξ . The same procedures that led to Eq. 24 would apply with (disorder-averaged) renormalized quantities.

Application to single-mode cavity strongly coupled to reactant subensemble. In the next section, we apply our theory to a reactive mixture where a single component of the molecular system strongly interacts with a microcavity represented by a single boson mode. This is a highly idealized scenario relative to most experiments for the reasons that we indicated above, e.g., polyatomic molecules have multiple bright vibrational modes and a continuous set of on and off-resonant EM modes contribute to the observation of polariton effects on chemical equilibria in planar cavities. Nevertheless, this remains a nonperturbative treatment of strong light-matter interaction effects which follow from the formation of hybrid (polariton) modes in the confined environment. Further, as we demonstrate below, our analysis of polaritonic effects on equilibria in single-mode EM resonators already indicates several important qualitative trends that are expected to persist in any complete treatment including a macroscopic number of molecular and EM degrees of freedom.

In the case where only reactant species A strongly interacts with a single EM mode and the number of molecules of type A (obtained from solving Eq. 24) is N_A , the nonperturbative light-matter Hamiltonian contains $N_A + 1$ eigenmodes corresponding to the $N_A - 1$ purely molecular modes that have the same spectrum as the bright vibrations of A and the hybrid LP and UP. The contribution of the $N_A - 1$ reservoir normal modes to $\mu_{F,Pol}$ cancels out the term $\mu_{A,vib,bright}(T)$ in $\Delta\mu_{A,Pol}(N, V, T)$ (Eq. 21). As expected, the effect of nonperturbative light-matter interactions on the composition of the reactive mixture at equilibrium in this example is entirely due to the isolated LP and UP modes. Let the polariton effect on chemical equilibrium $F_{Pol}(V, T)$ be defined as the ratio of the equilibrium reaction quotient inside the microcavity $R(V, T) = N_E^{\nu_E} N_C^{\nu_C} / N_A^{\nu_A} N_B^{\nu_B}$ to the standard-state reaction quotient (equilibrium constant) $K^{(0)}(T) = (q_E^{\nu_E} q_C^{\nu_C}) / (q_A^{\nu_A} q_B^{\nu_B})$ (assuming ideal-gas conditions for simplicity). It follows from Eq. 24 that at equilibrium the polariton effect on the reaction quotient is given by

$$F_{Pol}(V, T) = \frac{R(V, T)}{K^{(0)}(T)} = e^{-\beta\nu_A [\mu_A^{LP}(N_A, V, T) + \mu_A^{UP}(N_A, V, T)]}, \quad (25)$$

where N_A is obtained by solving the equation $R(V, T) = K^{(0)}(T) e^{-\beta\nu_A [\mu_A^{LP}(N_A, V, T) + \mu_A^{UP}(N_A, V, T)]}$, and the changes in the chemical potential of the A molecules due to LP and UP are given by

$$\mu_A^{LP}(N_A, V, T) = \frac{\partial A_{LP}(N_A, V, T)}{\partial N_A}, \quad (26)$$

where $A_{LP} = -k_B T \ln q_{LP}(N_A, V, T)$, and identical definitions exist for UP. This equation forms the basis for

the qualitative and quantitative analysis of a model gas-phase bimolecular nucleophilic substitution reaction that we discuss in the next section.

Note that, as is well known [58–60], the degeneracy of the $N_F - 1$ dark modes is easily broken as they become weakly coupled to light in the presence of molecular permutational-symmetry breaking perturbations. This does not change Eq. 25 in any appreciable way, since the difference between the weakly coupled reservoir density of states and that of the molecular system in free space is negligible in the collective strong light-matter interaction regime of interest to us [60, 61]. Therefore, the same cancellation between the free space bright vibrational contribution to the chemical potential of the F subensemble and the molecular dark reservoir inside an optical cavity occurs to a large extent when the number of molecules is sufficiently large, i.e., when $N_F \rightarrow \infty$, and thus the results obtained in the presence of permutational symmetry $\Delta\mu_{F,\text{Pol}} = \mu_F^{\text{LP}} + \mu_F^{\text{LP}}$ remain a very good approximation for a molecular ensemble interacting with a single boson mode.

III. BIMOLECULAR NUCLEOPHILIC SUBSTITUTION REACTION IN A SINGLE-MODE CAVITY

A. Model

To illustrate the theory described above, we consider a lossless single-mode cavity interacting with a gas-phase reactive mixture where equilibrium is established via the S_N2 reaction

This reaction has been thoroughly studied in the gas phase [62, 63]. We construct its chemical equilibrium constant in free space from the gas-phase partition function of each chemical species assuming separability between the internal degrees of freedom and ideal gas conditions. We ignore the electronic contribution to the reaction free energy, for in this case, the equilibrium constant is close to 1 at room temperature and strong light-matter coupling becomes more likely to significantly affect the composition of the reactive mixture at equilibrium. Vibrational contributions to the partition function were constructed using the quantum harmonic oscillator model, whereas classical partition functions were employed in the treatment of rotational and translational degrees of freedom of each species (see Appendix 1). Normal-mode frequencies and moments of inertia required for the partition functions of the chemical species in Eq. 27 were extracted from data available in the Chemistry WebBook [64].

In order to probe polariton effects on the chemical equilibrium associated with Eq. 27, we suppose the system is embedded in an optical cavity with a single high-quality mode in resonance with a particular vibrational mode of

reactants or products. To examine the distinct effects of reactant and product strong light-matter coupling, we chose two strongly absorbing IR modes of reactants and products in the gas phase [64]. The frequencies of the selected dipole-active normal modes are given in Table I.

When a single normal mode (of N_F reactant or product molecules) interacts nonperturbatively with the optical microcavity, the polaritonic part of the Coulomb gauge [38] light-matter Hamiltonian can be written in the uncoupled basis as [7]

$$H = \sum_{i=1}^{N_F} \hbar\omega_F a_i^\dagger a_i + \hbar\tilde{\omega}_C b^\dagger b - ig \sqrt{\frac{\omega_M}{\tilde{\omega}_C}} \sum_{j=1}^{N_F} (a_j^\dagger - a_j)(b^\dagger + b), \quad (28)$$

where ω_F is the frequency of the strongly coupled molecular normal mode (of type F), a_i^\dagger and a_i are the creation and annihilation operators of F excitations in the i th molecule, and b^\dagger and b are the creation and annihilation operators of the cavity mode with renormalized frequency $\tilde{\omega}_C$ given by

$$\tilde{\omega}_C^2 = \omega_C^2 + \Omega_R^2, \quad (29)$$

where ω_C is the bare photon frequency and $\Omega_R = 2g\sqrt{N_F}$ is the collective light-matter interaction strength. We have assumed throughout that the bare cavity mode is on resonance with a reactant or product normal mode (Table 1), therefore, we set $\omega_C = \omega_F$ from now on. When the renormalized cavity mode frequency is near-resonant with the molecular normal mode, the effective collective light-matter interaction strength for the strongly coupled species is $g\sqrt{N\omega_F/\tilde{\omega}_C} \approx g\sqrt{N_F}$. Note that the reactant and product modes listed in Table I have essentially equal oscillator strength [64], and, therefore we will employ the same value of g when analyzing the effects on chemical equilibrium induced by exclusive strong light-matter coupling with each mode.

The light-matter system described by the Hamiltonian given by Eq. 28 has $N_F + 1$ eigenmodes. The frequencies of the hybrid excitations (polaritons) are

$$\omega_{\text{LP}} = \sqrt{\frac{\tilde{\omega}_C^2 + \omega_F^2 - \sqrt{(\tilde{\omega}_C^2 - \omega_F^2)^2 + 4\Omega_R^2\omega_F^2}}{2}}, \quad (30)$$

$$\omega_{\text{UP}} = \sqrt{\frac{\tilde{\omega}_C^2 + \omega_F^2 + \sqrt{(\tilde{\omega}_C^2 - \omega_F^2)^2 + 4\Omega_R^2\omega_F^2}}{2}}. \quad (31)$$

The remaining $N_F - 1$ normal-modes form a degenerate purely molecular reservoir with the same frequency ω_F as the bare molecules.

Using basic statistical mechanics [1, 2], we can obtain the polariton and reservoir mode partition functions and compute the polariton effect on the chemical potential $\Delta\mu_{F,\text{Pol}}$ (Eq:21) required to set up the nonlinear Eq. 24. Its solution consists of the equilibrium number of

P mode	ω (cm ⁻¹)	T_ν (K)	R mode	ω (cm ⁻¹)	T_ν (K)
CC Str	974	1402	CC Str	964	1388
CCl Str	677	974	CBr Str	583	839

TABLE I: Selected IR-active vibrational modes of C₂H₅Cl (P modes) and C₂H₅Br (R modes) with ω obtained from Ref. [66] and vibrational temperatures $T_\nu = h\nu/k_B$, where ν is the frequency of each mode.

molecules of each species inside the optical cavity, and allows us to establish the polariton effect on the chemical equilibrium as measured by $F_{\text{Pol}}(V, T)$ via Eq. 25.

The numerical problem is set up by assuming that the mixture initially contains an equal number of ethyl bromide and chloride ions $N = N_{\text{C}_2\text{H}_5\text{Br}}^0 = N_{\text{Cl}^-}^0$ that react according to Eq. 27 to establish equilibrium with ethyl chloride and bromine ions. The number of reactant and product molecules at equilibrium is denoted N_R and N_P respectively. It follows that at equilibrium $N_R = N_{\text{C}_2\text{H}_5\text{Br}} = N_{\text{Cl}^-}$ and $N_P = N_{\text{C}_2\text{H}_5\text{Cl}} = N_{\text{Br}^-} = N - N_R$. The standard-state equilibrium constant $K_0(T)$ (outside the microcavity) is computed as a function of temperature using the ratio of product and reactant partition functions. To find the equilibrium composition of the mixture at thermal equilibrium, we solve the equation

$$\frac{(N - N_R)^2}{N_R^2} = K_0(T) e^{\beta \nu_F [\mu_F^{\text{LP}}(N_F, V, T) + \mu_F^{\text{UP}}(N_F, V, T)]}, \quad (32)$$

for N_R , where ν_F is the stoichiometric coefficient of the strongly coupled species, and $N_F = N_R$ or N_P when strong coupling occurs with a reactant or product normal mode, respectively. We solve Eq. 32 for a given T , initial number of molecules N , and single-molecule light-matter interaction strength g . The standard-state equilibrium composition of the reactive mixture is employed as an initial guess for the solution, and the polariton contributions to the chemical potential are obtained from automatic differentiation of the polariton free energies with respect to the number of strongly coupled molecules as implemented in the python AutoGrad package [65].

In the case where a reactant mode strongly interacts with the microcavity, the polariton effect on the equilibrium composition of the reactive mixture can be written as a ratio of the equilibrium constant inside the cavity to the bare equilibrium constant (Eq. 17)

$$F_{\text{Pol}}^{\text{R}} = \exp[-\beta \mu_{\text{R,Pol}}(N_R)], \quad (33)$$

where $\mu_{\text{R,Pol}}(N_R) = \mu_{\text{R}}^{\text{LP}}(N_R) + \mu_{\text{R}}^{\text{UP}}(N_R)$. Conversely, if strong coupling occurs with the product molecules, we obtain

$$F_{\text{Pol}}^{\text{P}} = \exp[\beta \mu_{\text{P,Pol}}(N_P)]. \quad (34)$$

B. Results and Discussion

We have investigated the effect of single-mode strong light-matter coupling on the equilibrium composition of the reactive molecular mixture described by Eq. 27 at various temperatures, system sizes, and light-matter interaction strengths assuming that strong coupling occurs between the cavity and a single set of normal vibrational modes of reactant or product.

The bare cavity frequency ω_C is set to be on resonance with the strongly coupled vibrational mode. Renormalization (Eq. 29) of the cavity frequency in the presence of the molecular system leads to a nonzero detuning that is insignificant relative to the light-matter interaction strength under the conditions analyzed in this work.

The temperature dependence of the ratio between the reaction quotient of the selected S_N2 reaction inside and outside a microcavity is provided in Fig. 2. This figure shows four notable features: a. polariton effects are strongest at low temperatures and vanish at the high-temperature limit, b. the equilibrium is shifted towards the products when reactants are strongly coupled to light and vice-versa, c. the observed effects are extremely small considering the large single-molecule light-matter coupling strength employed (for the purposes of illustrating our theory), and d. polaritons formed between molecular modes with lower frequency have a stronger impact on the chemical equilibrium constant. Below, we discuss each of these trends.

FIG. 2: Temperature dependence of the single-mode cavity effect on the examined S_N2 equilibrium. The single-molecule light-matter coupling strength is $g = 10$ cm⁻¹ and the maximum number of strongly coupled modes is $N = 100$.

Low- and high-temperature behavior. Figure 2 shows the single-mode cavity effect on the composition of the equilibrium reactive mixture is largest at low temperatures, whereas strong coupling has no effect in the high-temperature limit. To understand this, note that at low temperatures, the polaritons and bare modes are essentially in their ground-state, and therefore any polariton-induced change in free energy responsible for modifying chemical equilibrium is generated by the difference between polariton and bare molecule zero-point energies. At high temperatures, the classical limit of the light-

FIG. 3: Polariton-induced changes in free energy, internal energy and entropic contribution to the free energy per unit interacting degree of freedom (Δa , Δe , and $T\Delta s$, respectively) as a function of temperature in cm^{-1} units. The left figure shows the results obtained when the CC Str mode of ethyl bromide is strongly coupled to the cavity, with $g = 10 \text{ cm}^{-1}$ and $N = 100$. The right figure shows analogous results when the CBr Str mode is strongly coupled to the cavity. These figures reveal that polariton-induced changes in the system's zero-point energy provide the dominant contribution to the temperature-dependent single-mode VSC effect on chemical equilibrium reported in Fig. 2, and that entropic changes become relevant as T grows from ultracold ($T < 100 \text{ K}$) to the vibrational temperature $\hbar\nu/k_B$ of the strongly coupled mode with frequency ν . The change in internal energy Δe per molecule is an increasing function of the temperature at low T because the LP mode frequency is lower than the bare molecular, and thus LP has greater likelihood to thermally occupy the state with a single quantum than the bare molecular vibration at $T \ll T_\nu$. This same variation of occupation number explains the observed polariton-induced increase in entropy at low T due to VSC. The computed effects are seen to be small, but the quantitative details shown here strictly apply only to the case where a single cavity-mode interacts with a single normal-mode of the reactant or product ensemble.

matter partition function can be employed to show that the free energy of the reactive mixture is unaffected by polariton formation[35, 36].

In Fig. 3, we examine the polariton-induced variation with temperature of the change per strongly coupled degree of freedom (molecular and photonic) in the internal energy $\Delta e = \Delta E/(N_F + 1)$, free energy $\Delta a = \Delta A/(N_F + 1)$ and $T\Delta s = T\Delta S/(N_F + 1)$ of the system at equilibrium

$$\Delta e = \frac{E_{LP} + E_{UP} - E_F - E_C}{N_F + 1}, \quad (35)$$

$$\Delta a = \frac{A_{LP} + A_{UP} - A_F - A_C}{N_F + 1}, \quad (36)$$

$$T\Delta s = \frac{S_{LP} + S_{UP} - S_F - S_C}{N_F + 1}, \quad (37)$$

where F is either R or P and N_F is the number of strongly coupled molecules at equilibrium. We limit our discussion to strong coupling with the reactant ensemble ($\text{C}_2\text{H}_5\text{Br}$) since the conclusions we derive here are straightforwardly generalizable to the case where strong coupling occurs with products.

FIG. 4: Schematic representation of single-mode VSC effect on chemical equilibria via changes in the reaction free energy, due to different light-matter interaction of reactants and products. The figure illustrates the case where reactants are strongly coupled to the considered cavity mode. In this case, the reactant free energy is effectively raised, thus leading to a reduction in the reaction free energy, and a field-induced shift of the chemical equilibrium towards the products.

Fig. 3 shows the observed polariton effect in the reactive mixture composition (Fig. 2) at low T is essentially due to the cavity-induced change in reactant or product zero-point energies. This follows from the fact that at the low- T limit, Δe is entirely determined by the zero-point energy of the degrees of freedom involved in strong light-matter coupling

$$\lim_{T \rightarrow 0} \Delta e = \frac{\hbar(\omega_{LP} + \omega_{UP} - \omega_F - \omega_C)}{2(N_F + 1)}. \quad (38)$$

Conversely, the entropy contribution of all modes vanish as $T \rightarrow 0$. Therefore, it follows, given that $\omega_{LP} + \omega_{UP} - \omega_F - \omega_C \neq 0$, the change in system free energy induced by the optical cavity at low temperatures relative to the vibrational temperature of the strongly coupled modes is dominated by the ground-state energy difference between the polariton normal-modes and the microcavity and molecular vibrational modes.

At higher temperatures, the polariton effect on the composition of the molecular mixture at thermodynamic equilibrium becomes negligible ($F_{Pol} \rightarrow 1$) regardless of the vibrational frequency and light-matter coupling strength. The absence of any effect on the internal energy may be seen from the equipartition theorem (this implies that each normal mode has $k_B T$ mean internal energy) [1, 2], whereas entropy variations induced by strong light-matter coupling may be seen to vanish from the classical limit of the harmonic oscillator partition functions which

can be directly applied to give

$$\begin{aligned} \lim_{T \rightarrow \infty} \Delta s &\approx \frac{1}{N_F + 1} \ln \left(\frac{q_{LP} q_{UP}}{q_F q_C} \right) \\ &= \frac{1}{2(N_F + 1)} \ln \left(\frac{\omega_C^2}{\omega_C^2 + \Omega_R^2 - \Omega_R^2} \right) = 0. \end{aligned} \quad (39)$$

It follows that $T\Delta s$ goes to 0 at low and high T but is an increasing function of T at intermediate temperatures, therefore showing a maximum (Fig. 3).

Note the quantum treatment of field and molecular vibrational modes is essential, as the polariton-induced change in molecular free energy at the experimentally relevant temperatures $T \ll T_\nu$ is dictated by the Bose-Einstein distribution. Quantum statistics is relevant here, for the contribution of excited-states to vibrational free energies is essentially negligible for high-frequency normal-modes with $T_\nu \gg T$, and exactly under such conditions, the Bose-Einstein distribution is significantly different from the classical Maxwell-Boltzmann. This explains the distinct polariton effects on thermodynamic properties of molecular systems observed here relative to those examined in the classical limit by Li et al [35].

Direction of chemical equilibrium shift. Figure 3 also explains why single-mode strong coupling with a chemical species tends to bias the equilibrium towards the uncoupled species. This occurs because the sum of polariton zero-point energies $E_{LP} + E_{UP}$ is greater than the sum of the bare molecule normal-mode and bare photon zero-point energies. This feature increases the free energy of the light-matter system inside the microcavity relative to the bare system.

Magnitude of polariton effect on chemical equilibrium. The single-mode strong coupling effect on chemical equilibrium as measured by the ratio of the reaction quotient in the microcavity to the standard-state (bare) equilibrium constant is observed to be less than 1.003 even at low temperatures such as 50 K. The effect becomes even weaker at higher temperatures, and may be understood from Fig. 3, which shows that the (single-cavity mode) polariton effect on the free energy *per degree of freedom* is extremely small. We revisit this point when discussing the system-size dependence of our results later. Note the quantitative results present in this section do not rule out a polaritonic effect on chemical equilibria in complex EM environments where multiple vibrational modes of the reactive species strongly interact with a continuous set of on and off-resonance cavity modes as in a planar microcavity.

Strong coupling with lower frequency normal-modes have greater impact on chemical equilibrium constants. Fig. 2 shows that strong light-matter coupling is most effective (among the scenarios we considered) when the matter part of polaritons corresponds to the CCl (product normal-mode) or CBr (reactant normal-mode) stretch modes. These motions have lower frequency than the CC stretch of either reactants or products by about 300 and 400 cm^{-1} , respectively. The greater impact

FIG. 5: (a) Polariton effect on chemical equilibrium constant vs single-molecule light-matter coupling strength for a system with a maximum number of 100 product molecules at $T = 300$ K. (b) Polariton effect on equilibrium at $T = 300$ K and $g = 10 \text{ cm}^{-1}$ vs total number of molecules inside cavity.

of VSC occurring with lower frequency vibrations may be understood mathematically from an analysis of the polariton contribution to the zero-point energy difference between the polaritonic system and the composite (light-matter) bare system per (strongly coupled) degree of freedom. Under the conditions examined here where $\tilde{\omega}_C \approx \omega_M$ and $g/\omega_M \ll 1$, the polariton effect at the zero-point energy difference is given by

$$\lim_{T \rightarrow 0} \Delta e \approx \frac{g^2}{2\omega_M}, \quad g/\omega_M \ll 1. \quad (40)$$

This result clearly demonstrates that at low temperatures, where single-mode cavity effects on equilibrium are largest, light-matter interactions will have more significant impact when it involves lower frequency and stronger oscillator strength vibrational modes [36]. Similar results are valid at higher temperatures where thermal excitations play a greater role but ultimately lead to no polariton effect at chemical equilibrium in the $T \rightarrow \infty$ limit [36].

Size and oscillator strength dependence of polariton effects on chemical equilibrium. In the examined reaction, all strongly coupled modes have nearly equal oscillator strength. However, this is not a generic feature of polyatomic molecules, which will generally have vibrational excitations with variable absorption intensity. In order to assess the dependence of F_{Pol} on the single-molecule light-matter coupling strength, we present in Fig. 5(a) the behavior of F_{Pol} at $T = 300$ K as a function of the single-molecule light-matter coupling constant. As expected, the polariton effect on the composition of the reactive mixture is enhanced with increasing single-molecule light-matter coupling strength. This behavior follows expectations based on Eq. 40 and others similarly related [36].

We end our analysis of single-mode microcavity effects on chemical equilibrium by quantitatively investigating the behavior of the polariton effect under changes in the maximum number of strongly coupled molecules N (with

fixed cavity volume). We find that while Ω_R increases, the overall VSC effect on the reactive mixture composition decreases substantially as the number of molecules increases at $T = 300\text{K}$ [Fig. 5(b)].

The weakening of VSC-induced changes on chemical equilibrium constants with increasing molecular density is an expected feature of single-cavity mode theories [35, 67, 68] which have systematically shown that polariton effects on local molecular observables decrease with increasing molecular density. Fig. 5 shows a substantial deviation of the scaling of the strong coupling effect on the examined chemical equilibrium constant relative to simple $1/N$ scaling. Still, the implication of various earlier studies [35, 67, 69–71] remain valid that single-mode cavity effects on local molecular observables are insignificant in the collective light-matter interaction regime.

IV. CONCLUSIONS

We provided a general theory of chemical equilibrium under nonperturbative light-matter interactions and applied it to an S_N2 reaction in a single-mode cavity. We found that polaritons can shift chemical equilibrium constants towards either direction of a reaction depending on the species (reactant or product) strongly coupled to the EM field, light-matter interaction effects are most potent at lower temperatures, and the change induced by VSC on the internal energy of the light-matter system provide the dominant contribution to the VSC effect on chemical equilibria. We also showed that strong light-matter coupling is more effective at shifting chemical equilibria when polaritons are formed between IR cavity modes and molecular vibrations with more significant oscillator strength and lower frequency.

These trends were obtained in an idealized scenario where strong light-matter coupling occurs between a single EM mode of an IR resonator and the normal modes of a particular component of the reactive mixture (reactant or product ensemble) but are based on fundamental features of our theory that are expected to hold more generally. Future work, based on Eq. 24 and discussed generalizations accounting for intermolecular interactions and including a quantitative analysis of

equilibrium effects in realistic systems with simultaneous strong coupling of multiple IR modes of reactants and products with multimode Fabry-Perot cavities will allow direct comparison with experiments [3].

Data availability. The data that support the findings of this study are available from the corresponding author upon reasonable request.

ACKNOWLEDGMENTS

RFR acknowledges generous startup funds from Emory University.

APPENDIX A. Partition functions

In this Appendix, we provide explicit forms for the partition functions we employed to obtain the polariton effect on the chemical equilibrium described by Eq. 27. In particular, the translational, (classical) rotational and (quantum) vibrational partition functions of an asymmetric top molecule with n normal-modes were obtained from Ref. [2]

$$q_{\text{trans}}(V, T) = \left(\frac{2\pi m k_B T}{h^2} \right)^{3/2} V, \quad (41)$$

$$q_{\text{rot}}(T) = \frac{\pi^{1/2}}{\sigma} \prod_{j=1}^3 \left(\frac{8\pi^2 I_j k_B T}{h^2} \right)^{1/2} \quad (42)$$

$$q_{\text{vib}}(T) = \prod_{a=1}^n \frac{e^{-\beta \hbar \omega_a / 2}}{1 - e^{-\beta \hbar \omega_a}}, \quad (43)$$

where m is the molecular mass, V is the volume occupied by the system, I_1, I_2 and I_3 denote principal moments of inertia, σ is the molecule's symmetry number, and ω_a are normal-mode frequencies. Note that we only worked with temperatures that are much lower than the electronic energy gaps for each molecule. Thus, only the electronic ground-state of each species is occupied.

V. REFERENCES

-
- [1] D. A. McQuarrie, *Statistical mechanics* (Sterling Publishing Company, 2000).
- [2] R. K. Pathria, *Statistical mechanics* (Elsevier, 2016).
- [3] Y. Pang, A. Thomas, K. Nagarajan, R. M. A. Vergauwe, K. Joseph, B. Patraha, K. Wang, C. Genet, and T. W. Ebbesen, On the role of symmetry in vibrational strong coupling: The case of charge-transfer complexation, *Angewandte Chemie International Edition* **59**, 10436 (2020).
- [4] T. W. Ebbesen, Hybrid light-matter states in a molecular and material science perspective, *Accounts of chemical research* **49**, 2403 (2016).
- [5] R. F. Ribeiro, L. A. Martínez-Martínez, M. Du, J. Campos-Gonzalez-Angulo, and J. Yuen-Zhou, Polariton chemistry: Controlling molecular dynamics with optical cavities, *Chemical Science* **9**, 6325–6339 (2018).
- [6] M. Hertzog, M. Wang, J. Mony, and K. Börjesson, Strong light-matter interactions: a new direction within chemistry, *Chemical Society Reviews* **48**, 937 (2019).
- [7] A. V. Kavokin, J. J. Baumberg, G. Malpuech, and F. P. Laussy, *Microcavities*, Vol. 21 (Oxford university press, 2017).

- [8] K. J. Vahala, Optical microcavities, *nature* **424**, 839 (2003).
- [9] J. Bellessa, C. Bonnard, J. Plenet, and J. Mugnier, Strong coupling between surface plasmons and excitons in an organic semiconductor, *Physical review letters* **93**, 036404 (2004).
- [10] T. Hakala, J. Toppari, A. Kuzyk, M. Pettersson, H. Tikkanen, H. Kunttu, and P. Törmä, Vacuum rabi splitting and strong-coupling dynamics for surface-plasmon polaritons and rhodamine 6g molecules, *Physical review letters* **103**, 053602 (2009).
- [11] R. Chikkaraddy, B. De Nijs, F. Benz, S. J. Barrow, O. A. Scherman, E. Rosta, A. Demetriadou, P. Fox, O. Hess, and J. J. Baumberg, Single-molecule strong coupling at room temperature in plasmonic nanocavities, *Nature* **535**, 127 (2016).
- [12] A. Thomas, J. George, A. Shalabney, M. Dryzhakov, S. J. Varma, J. Moran, T. Chervy, X. Zhong, E. Devaux, C. Genet, *et al.*, Ground-state chemical reactivity under vibrational coupling to the vacuum electromagnetic field, *Angewandte Chemie* **128**, 11634 (2016).
- [13] J. Lather, P. Bhatt, A. Thomas, T. W. Ebbesen, and J. George, Cavity catalysis by cooperative vibrational strong coupling of reactant and solvent molecules, *Angewandte Chemie International Edition* **58**, 10635 (2019).
- [14] A. Thomas, L. Lethuillier-Karl, K. Nagarajan, R. M. Vergauwe, J. George, T. Chervy, A. Shalabney, E. Devaux, C. Genet, J. Moran, *et al.*, Tilting a ground-state reactivity landscape by vibrational strong coupling, *Science* **363**, 615 (2019).
- [15] K. Hirai, R. Takeda, J. A. Hutchison, and H. Uji-i, Modulation of prins cyclization by vibrational strong coupling, *Angewandte Chemie* **132**, 5370 (2020).
- [16] A. Thomas, A. Jayachandran, L. Lethuillier-Karl, R. M. Vergauwe, K. Nagarajan, E. Devaux, C. Genet, J. Moran, and T. W. Ebbesen, Ground state chemistry under vibrational strong coupling: dependence of thermodynamic parameters on the rabi splitting energy, *Nanophotonics* **9**, 249 (2020).
- [17] J. Lather and J. George, Improving enzyme catalytic efficiency by co-operative vibrational strong coupling of water, *The Journal of Physical Chemistry Letters* **12**, 379 (2020).
- [18] A. Sau, K. Nagarajan, B. Patrahau, L. Lethuillier-Karl, R. M. Vergauwe, A. Thomas, J. Moran, C. Genet, and T. W. Ebbesen, Modifying woodward–hoffmann stereoselectivity under vibrational strong coupling, *Angewandte Chemie International Edition* **60**, 5712 (2021).
- [19] K. Sandeep, K. Joseph, J. Gautier, K. Nagarajan, M. Sujith, K. G. Thomas, and T. W. Ebbesen, Manipulating the self-assembly of phenyleneethynylenes under vibrational strong coupling, *The Journal of Physical Chemistry Letters* **13**, 1209 (2022).
- [20] K. Nagarajan, J. George, A. Thomas, E. Devaux, T. Chervy, S. Azzini, K. Joseph, A. Jouaiti, M. W. Hosseini, A. Kumar, C. Genet, N. Bartolo, C. Ciuti, and T. W. Ebbesen, Conductivity and photoconductivity of a p-type organic semiconductor under ultrastrong coupling, *ACS Nano* **14**, 10219 (2020).
- [21] E. Orgiu, J. George, J. A. Hutchison, E. Devaux, J. F. Dayen, B. Doudin, F. Stellacci, C. Genet, J. Schachenmayer, C. Genes, and *et al.*, Conductivity in organic semiconductors hybridized with the vacuum field, *Nature Materials* **14**, 1123–1129 (2015).
- [22] T. Fukushima, S. Yoshimitsu, and K. Murakoshi, Inherent promotion of ionic conductivity via collective vibrational strong coupling of water with the vacuum electromagnetic field, *Journal of the American Chemical Society* **144**, 12177 (2022).
- [23] X. Zhong, T. Chervy, L. Zhang, A. Thomas, J. George, C. Genet, J. A. Hutchison, and T. W. Ebbesen, Energy transfer between spatially separated entangled molecules, *Angewandte Chemie International Edition* **56**, 9034 (2017).
- [24] K. Akulov, D. Bochman, A. Golombek, and T. Schwartz, Long-distance resonant energy transfer mediated by hybrid plasmonic–photonic modes, *The Journal of Physical Chemistry C* **122**, 15853 (2018).
- [25] X. Zhong, T. Chervy, S. Wang, J. George, A. Thomas, J. A. Hutchison, E. Devaux, C. Genet, and T. W. Ebbesen, Non-radiative energy transfer mediated by hybrid light-matter states, *Angewandte Chemie International Edition* **55**, 6202 (2016).
- [26] D. M. Coles, N. Somaschi, P. Michetti, C. Clark, P. G. Lagoudakis, P. G. Savvidis, and D. G. Lidzey, Polariton-mediated energy transfer between organic dyes in a strongly coupled optical microcavity, *Nature Materials* **13**, 712–719 (2014).
- [27] K. Georgiou, P. Michetti, L. Gai, M. Cavazzini, Z. Shen, and D. G. Lidzey, Control over energy transfer between fluorescent bodipy dyes in a strongly coupled microcavity, *ACS Photonics* **5**, 258 (2018).
- [28] M. Du and J. Yuen-Zhou, Catalysis by dark states in vibropolaritonic chemistry, *Phys. Rev. Lett.* **128**, 096001 (2022).
- [29] X. Li, A. Mandal, and P. Huo, Cavity frequency-dependent theory for vibrational polariton chemistry, *Nature Communications* **12** (2021).
- [30] E. W. Fischer, J. Anders, and P. Saalfrank, Cavity-altered thermal isomerization rates and dynamical resonant localization in vibro-polaritonic chemistry, *The Journal of Chemical Physics* **156**, 154305 (2022).
- [31] D. S. Wang, T. Neuman, S. F. Yelin, and J. Flick, Cavity-modified unimolecular dissociation reactions via intramolecular vibrational energy redistribution, *The Journal of Physical Chemistry Letters* **13**, 3317–3324 (2022).
- [32] J. Sun and O. Vendrell, Suppression and enhancement of thermal chemical rates in a cavity, *The Journal of Physical Chemistry Letters* **13**, 4441–4446 (2022).
- [33] P.-Y. Yang and J. Cao, Quantum effects in chemical reactions under polaritonic vibrational strong coupling, *The Journal of Physical Chemistry Letters* **12**, 9531 (2021).
- [34] G. D. Scholes, C. A. DelPo, and B. Kudisch, Entropy reorders polariton states, *The Journal of Physical Chemistry Letters* **11**, 6389 (2020).
- [35] T. E. Li, A. Nitzan, and J. E. Subotnik, On the origin of ground-state vacuum-field catalysis: Equilibrium consideration, *The Journal of chemical physics* **152**, 234107 (2020).
- [36] P. Pilar, D. De Bernardis, and P. Rabl, Thermodynamics of ultrastrongly coupled light-matter systems, *Quantum* **4**, 335 (2020).
- [37] Y. Ashida, A. İmamoglu, J. Faist, D. Jaksch, A. Cavalleri, and E. Demler, Quantum electrodynamic control of matter: Cavity-enhanced ferroelectric phase transition, *Physical Review X* **10**, 041027 (2020).

- [38] D. P. Craig and T. Thirunamachandran, *Molecular quantum electrodynamics: an introduction to radiation-molecule interactions* (Courier Corporation, 1998).
- [39] J. D. Jackson, *Classical electrodynamics* (John Wiley & Sons, 2021).
- [40] G. J. Aroeira, K. T. Kairys, and R. F. Ribeiro, Theoretical analysis of exciton wave packet dynamics in polaritonic wires, *The Journal of Physical Chemistry Letters* **14**, 5681 (2023).
- [41] J. P. Long and B. Simpkins, Coherent coupling between a molecular vibration and fabry-perot optical cavity to give hybridized states in the strong coupling limit, *ACS photonics* **2**, 130 (2015).
- [42] A. Shalabney, J. George, J. a. Hutchison, G. Pupillo, C. Genet, and T. W. Ebbesen, Coherent coupling of molecular resonators with a microcavity mode, *Nature communications* **6**, 5981 (2015).
- [43] S. R. Casey and J. R. Sparks, Vibrational strong coupling of organometallic complexes, *The Journal of Physical Chemistry C* **120**, 28138 (2016).
- [44] A. Dunkelberger, B. Spann, K. Fears, B. Simpkins, and J. Owrutsky, Modified relaxation dynamics and coherent energy exchange in coupled vibration-cavity polaritons, *Nature Communications* **7**, 13504 (2016).
- [45] A. D. Wright, J. C. Nelson, and M. L. Weichman, Rovibrational polaritons in gas-phase methane, *Journal of the American Chemical Society* **145**, 5982 (2023).
- [46] V. I. Arnol'd, *Mathematical methods of classical mechanics*, Vol. 60 (Springer Science & Business Media, 2013).
- [47] C. Eckart, Some studies concerning rotating axes and polyatomic molecules, *Physical Review* **47**, 552 (1935).
- [48] J. D. Louck and H. W. Galbraith, Eckart vectors, eckart frames, and polyatomic molecules, *Reviews of Modern Physics* **48**, 69 (1976).
- [49] E. B. Wilson, J. C. Decius, and P. C. Cross, *Molecular vibrations: the theory of infrared and Raman vibrational spectra* (Courier Corporation, 1980).
- [50] P. W. Atkins and R. S. Friedman, *Molecular quantum mechanics* (Oxford university press, 2011).
- [51] W. Thomas, Über die zahl der dispersionselektronen, die einem stationären zustande zugeordnet sind.(vorläufige mitteilung), *Naturwissenschaften* **13**, 627 (1925).
- [52] F. Reiche and W. Thomas, Über die zahl der dispersionselektronen, die einem stationären zustand zugeordnet sind, *Zeitschrift für Physik* **34**, 510 (1925).
- [53] W. Kuhn, Über die gesamtstärke der von einem zustande ausgehenden absorptionslinien, *Zeitschrift für Physik* **33**, 408 (1925).
- [54] T. E. Li, A. Nitzan, and J. E. Subotnik, Polariton relaxation under vibrational strong coupling: Comparing cavity molecular dynamics simulations against fermi's golden rule rate, *The Journal of Chemical Physics* **156** (2022).
- [55] T. E. Li, A. Nitzan, and J. E. Subotnik, Energy-efficient pathway for selectively exciting solute molecules to high vibrational states via solvent vibration-polariton pumping, *Nature Communications* **13**, 4203 (2022).
- [56] M. Litinskaya and P. Reineker, Loss of coherence of exciton polaritons in inhomogeneous organic microcavities, *Physical Review B* **74**, 165320 (2006).
- [57] G. Engelhardt and J. Cao, Polariton localization and dispersion properties of disordered quantum emitters in multimode microcavities, *Physical Review Letters* **130**, 213602 (2023).
- [58] T. Botzung, D. Hagenmüller, S. Schütz, J. Dubail, G. Pupillo, and J. Schachenmayer, Dark state semilocalization of quantum emitters in a cavity, *Physical Review B* **102**, 144202 (2020).
- [59] G. D. Scholes, Polaritons and excitons: Hamiltonian design for enhanced coherence, *Proceedings of the Royal Society A* **476**, 20200278 (2020).
- [60] R. F. Ribeiro, Multimode polariton effects on molecular energy transport and spectral fluctuations, *Communications Chemistry* **5**, 48 (2022).
- [61] J. Dubail, T. Botzung, J. Schachenmayer, G. Pupillo, and D. Hagenmüller, Large random arrowhead matrices: Multifractality, semilocalization, and protected transport in disordered quantum spins coupled to a cavity, *Physical Review A* **105**, 023714 (2022).
- [62] C. Li, P. Ross, J. E. Szulejko, and T. B. McMahon, High-pressure mass spectrometric investigations of the potential energy surfaces of gas-phase sn2 reactions, *Journal of the American Chemical Society* **118**, 9360 (1996).
- [63] W. L. Hase, Simulations of gas-phase chemical reactions: applications to sn2 nucleophilic substitution, *Science* **266**, 998 (1994).
- [64] G. Caldwell, T. F. Magnera, and P. Kebarle, Sn2 reactions in the gas phase. temperature dependence of the rate constants and energies of the transition states. comparison with solution, *Journal of the American Chemical Society* **106**, 959 (1984).
- [65] D. Maclaurin, D. Duvenaud, and R. P. Adams, Autograd: Effortless gradients in numpy, in *ICML 2015 AutoML workshop*, Vol. 238 (2015).
- [66] P. J. Linstrom, *The NIST chemistry webbook* (distributed by American Institute of Chemical Engineers, 2002).
- [67] J. Galego, F. J. Garcia-Vidal, and J. Feist, Cavity-induced modifications of molecular structure in the strong-coupling regime, *Physical Review X* **5**, 041022 (2015).
- [68] R. F. Ribeiro and J. Yuen-Zhou, Introduction to vibropolaritons: Spectroscopy, relaxation and chemical reactions, in *Vibrational Dynamics of Molecules* (World Scientific, 2022) pp. 517–574.
- [69] J. A. Campos-Gonzalez-Angulo, R. F. Ribeiro, and J. Yuen-Zhou, Resonant catalysis of thermally activated chemical reactions with vibrational polaritons, *Nature communications* **10**, 4685 (2019).
- [70] V. P. Zhdanov, Vacuum field in a cavity, light-mediated vibrational coupling, and chemical reactivity, *Chemical Physics* **535**, 110767 (2020).
- [71] J. A. Campos-Gonzalez-Angulo and J. Yuen-Zhou, Polaritonic normal modes in transition state theory, *The Journal of chemical physics* **152**, 161101 (2020).

Reviewers' Comments:

Reviewer #1:

Remarks to the Author:

The authors addressed all my previous remarks and I am in favor of publishing the manuscript.

Reviewer #2:

Remarks to the Author:

I first want to thank the authors for their extensive explanations. This helped me to appreciate the main result, i.e. Eq.(24) in the current manuscript.

Despite this effort, however, it is far from obvious to me that the many assumptions the authors make in order to arrive at Eq.(24) are all justified. I will detail my reservations below point by point. On the other hand, if one accepts these assumptions and the order in which they are made, most of the steps seem reasonable. I will also give a list where more details and explanations seem necessary. Because my major concern is with respect to the authors' assumptions and I do not see any fundamental error in the derivations, I think the work can be published if the authors highlight the most important assumptions as detailed below. This would provide nice starting points for future research and connect to other communities.

Assumptions:

a. Disentangling the different degrees of freedom

The authors reference standard textbooks for free space, where there are no spatial restrictions and no collective coupling effects involved when disentangling translational, rotational, vibrational and electronic degrees of freedom. Without a detailed micro-canonical derivation it is not self-evident that all the standard assumptions valid for free space carry over to the cavity situation. For instance, already geometrical confinement leads to an adaptation of thermodynamics {1,2}. Besides, the usual considerations are always with respect to one molecule, whereas in the present case we need to take into account the state of the full ensemble and the potential inhomogeneity of the modes. I would ask the authors to highlight these assumptions and that there are interesting fundamental questions about the (quantum) statistics of a collectively coupled ensemble that can be explored.

In this context I would also not be so sure that the statement

"Note this partitioning of molecular degrees of freedom into classical and quantum does not pose any practical, nor conceptual challenge, especially as rotational and translational motions occur on timescales that are orders of magnitude slower than the vibrational dynamics involved in VSC."

is necessarily correct. The authors later assume that the approximate polaritonic modes are perfectly quantum coherent (see Eq.(11)) but parametrically dependent on all these degrees of freedom. This puts the polaritonic degrees of freedom on a similar level as electronic degrees of freedom in a Born-Oppenheimer approximation. The corresponding collective space (Θ, X) is extremely large and hence I would expect many degeneracies and non-adiabatic behavior. I think that such potential problems should be highlighted and not trivialized.

b. Quantum coherent polaritons and macroscopic averaging

In general I think it is necessary to highlight much more that the polaritonic modes of Eq.(11) are assumed quantum coherently delocalized over the full ensemble. Taking into account the potential large degeneracies and non-adiabatic behavior of the polaritonic modes I would expect strong decoherence and thus mostly classical behaviour emerging. Indeed, I would expect an analogous behaviour as for the total electronic wave function of the ensemble, which is usually assumed to decohere between the different molecules.

To some extent this seems assumed later when averaging over all rotational and translational degrees of freedom for Q_{pol} above Eq.(16). If I understand correctly the argument, the "mean-field frequencies" and the resulting Q_{pol} are essentially classical/macroscopic quantities. And I would expect that after averaging, the resulting distribution would be much more like a Maxwell-Boltzmann distribution, in contrast to the simplified model in Section III where no averaging procedure seems to be applied.

I would highlight that without any averaging procedure the microscopic polaritonic quantum modes do not become macroscopic and hence cannot compete with the otherwise macroscopic energy scaling of the rest of the system. In this case the effect of the polaritons diminishes with increasing system size. Hence I would also not be so sure that the difference between Ref.{3} and the current work (except of in this simplified model) is really due to the Bose-Einstein distribution.

c. Effective few-mode description

Following the above discussion, I think it is important to note that commonly the single- or few-mode approximation is a substitution for taking a part of the continuous photonic density of states into account effectively. Especially if one reduces to the mean-field polaritonic modes, I think it is more the macroscopicity of the molecular ensemble, as also stated in the current manuscript, that is responsible for any observable effect than the fact that a cavity contains a continuum of modes. I would propose to stress this a little more in the current manuscript.

Clarifications:

d. Parametric dependence and polaritonic eigenmodes

I think Eq.(10) is not rewriting the original Hamiltonian but approximating it for the subsequent thermal ensemble considerations. Clearly the Hamiltonian in Eq.(10) is not the same as the one of Eq.(2), which contains also all quantum states for rotations and translations. I also think it would be beneficial if the authors would write specifically in Eq.(11) the parametric dependence on all the coordinates. I think this would be helpful to distinguish the microscopic polaritonic degrees of freedom from the macroscopic ones upon averaging.

e. Configurational averaging

I am not so sure that the references the authors give for this procedure, i.e. [56,57] in the current manuscript, are the best ones, since they are for electronic strong coupling and based on perturbation theory. Could the authors show or discuss, maybe in an appendix, how such a procedure would be performed. In the end it seems everything relies on a reasonable approximation for the parametrically dependent frequencies and modes. I think it would be good to refer in this context to recent work on this topic {4,5}. This would explain better the mean-field frequencies which go into the final expression (24).

f. Definition of $q_F(V,T)$

In Eqs.(22-24) the expression $q_F(V,T)$ appears the first time, I believe. I think it is just the product of the different contributions as defined in Eq(15), but it would be good to unambiguously define all the quantities that appear in (24).

g. Discussion

I think to refer to N_p polaritonic modes and Eq.(11) is not correct, since Q_{pol} is the macroscopically averaged expression. I would again highlight that these averaged modes are macroscopic in nature and hence using a microscopic form will lead to negligible contributions as demonstrated in the example.

A point that would be good to address is the question of resonances. Is the formula (24) capable of capturing the pronounced resonance conditions as observed in experiments. Could the averaging and the parametrical dependence of the polariton modes generate something strongly peaked at a specific frequency? This would boil down to the question of how sensitive the polariton modes are to disordered in the micro-canonical ensemble.

h. Application to single-mode cavity

I think in this and the subsequent discussion it should be highlighted that there seems to be no macroscopic averaging of the polaritonic degrees of freedom. Thus it is unsurprising that for large number of molecules the results diminish. To me it seems that the authors take, instead of many micro-canonical realizations of the mean-field polariton, just one realization. Is this interpretation correct?

Minor points:

i. Maybe clarify that the use (k_x, k_y, k_z) is restricted to planar Fabry-Perot cavities, and for general cavity structures there is not necessarily a well-defined momentum associated.

j. Maybe indicate the upper limit in Eq.(7) and make p_α boldfaced below. Is it $\$N_A + \dots + N_E\$$? By the way, why E and not D in denoting the species?

k. In Eq.(6) and below (7) α refers to one particle in the Hamiltonian of species F (maybe indicate this). Later in Eq.(11) α refers to the microscopic polariton modes. Maybe choose a different notation.

l. Shouldn't the dependence in j_ζ^{ab} of Θ and Y be rather Θ_ζ and Y_ζ below Eq.(9)?

m. Maybe indicate somewhere explicitly that Θ , X , ... are actually multi-dimensional depending on the full ensemble of molecules, i.e., give explicitly the upper limit for ζ . Below Eq.(10) in H_{pol} there should also be a dependence on X . Why is there a ζ dependence on X in $V_M(\Theta, X)$?

n. Maybe indicate below Eq.(16) that the exponential in the definition of Q_{pol} is macroscopically averaged.

o. Above Eq.(18) the "A" of the free energy should not be a subindex. Maybe also briefly comment on how this chemical potential affects the photons, which are usually associated with zero chemical potential.

{1} Qiao, C. Z., et al. "Connect the thermodynamics of bulk and confined fluids: Confinement-adsorption scaling." *Langmuir* 35.10 (2019): 3840-3847.

{2} Dong, W., T. Franosch, and R. Schilling. "Thermodynamics, statistical mechanics and the vanishing pore width limit of confined fluids." *Communications Physics* 6.1 (2023): 161.

{3} Li, Tao E., Abraham Nitzan, and Joseph E. Subotnik. "On the origin of ground-state vacuum-field catalysis: Equilibrium consideration." *The Journal of chemical physics* 152.23 (2020).

{4} Schnappinger, Thomas, et al. "Cavity Born–Oppenheimer Hartree–Fock Ansatz: Light–Matter Properties of Strongly Coupled Molecular Ensembles." *The Journal of Physical Chemistry Letters* 14 (2023): 8024-8033.

{5} Schnappinger, Thomas, and Markus Kowalewski. "Ab-Initio Vibro-Polaritonic Spectra in Strongly Coupled Cavity-Molecule Systems." arXiv preprint arXiv:2310.01871 (2023).

Reviewer #3:

Remarks to the Author:

The authors have addressed all of my previous comments in this new version of the manuscript. I have no further comments. I recommend publication.

Response to reviewers of “Chemical equilibrium under vibrational strong coupling”

Kaihong Sun and Raphael F. Ribeiro
Department of Chemistry, Emory University
(Dated: December 2, 2023)

The reviews are provided below. Our response is given in blue and changes to the manuscript are in *italics*.

Reviewer 2.

I first want to thank the authors for their extensive explanations. This helped me to appreciate the main result, i.e. Eq.(24) in the current manuscript.

Despite this effort, however, it is far from obvious to me that the many assumptions the authors make in order to arrive at Eq.(24) are all justified. I will detail my reservations below point by point. On the other hand, if one accepts these assumptions and the order in which they are made, most of the steps seem reasonable. I will also give a list where more details and explanations seem necessary. Because my major concern is with respect to the authors’ assumptions and I do not see any fundamental error in the derivations, I think the work can be published if the authors highlight the most important assumptions as detailed below. This would provide nice starting points for future research and connect to other communities.

We thank the reviewer for the detailed reading of our work and address each point below.

Assumptions:

a. Disentangling the different degrees of freedom

The authors reference standard textbooks for free space, where there are no spatial restrictions and no collective coupling effects involved when disentangling translational, rotational, vibrational and electronic degrees of freedom. Without a detailed micro-canonical derivation it is not self-evident that all the standard assumptions valid for free space carry over to the cavity situation. For instance, already geometrical confinement leads to an adaptation of thermodynamics 1,2. Besides, the usual considerations are always with respect to one molecule, whereas in the present case we need to take into account the state of the full ensemble and the potential inhomogeneity of the modes. I would ask the authors to highlight these assumptions and that there are interesting fundamental questions about the (quantum) statistics of a collectively coupled ensemble that can be explored.

We thank the reviewer for bringing these points to our attention. We have further highlighted the separability assumption at new places in our manuscript which we list below for the convenience of the reviewer:

Page 4: *we infer (under the separability conditions and classical correspondence previously delineated), the thermodynamic properties of the total light-matter Hamiltonian ...*

Page 4: *Note the exclusion of nonadiabatic terms in Eq. 10 is consistent with the assumed separability of fast and slow molecular degrees of freedom.*

Page 11 (Conclusions): *Using separability conditions motivated from the disparate timescales of slow and fast molecular degrees of freedom, we obtained a nonlinear relation between equilibrium reaction quotients inside and outside a microcavity (Eq. 24) based on the polariton effect on the chemical potential of each component of the reactive mixture.*

We also added the following comments (page 2, right column) to the introduction, indicating that indeed there are some intriguing questions about quantum statistics of boson-fermion systems that we have neglected in this work .

A numerical investigation of the partition function associated to the many-body Hamiltonian in Eq. 2 in terms of its electronic-nuclear-photon stationary states would require a computationally unfeasible treatment of the intriguing mixed boson-fermion wave functions and the statistics of the light-matter system [refs]. Therefore, while our description has been general to this point, in what follows we specialize to the case where ...

In this context I would also not be so sure that the statement "Note this partitioning of molecular degrees of freedom into classical and quantum does not pose any practical, nor conceptual challenge, especially as rotational and translational motions occur on timescales that are orders of magnitude slower than the vibrational dynamics involved in VSC." is necessarily correct.

We understand the reviewer's skepticism in relationship to this comment as we have not provided a mathematically rigorous argument. We have changed the original text referred to by the reviewer to:

As we explain in more detail below, we expect this partitioning of molecular degrees of freedom to be reasonable based on the notion that rotations and translations occur on timescales that are much slower than the vibrational dynamics involved in VSC. Nevertheless, the interaction between fast polaritonic and slow matter degrees of freedom remains an intriguing issue for future work to unravel.

The authors later assume that the approximate polaritonic modes are perfectly quantum coherent (see Eq.(11)) but parametrically dependent on all these degrees of freedom. This puts the polaritonic degrees of freedom on a similar level as electronic degrees of freedom in a Born-Oppenheimer approximation. The corresponding collective space (Θ, \mathbf{X}) is extremely large and hence I would expect many degeneracies and non-adiabatic behavior. I think that such potential problems should be highlighted and not trivialized.

Regarding the matter of non-adiabatic effects, we have added the following text to page 4, which highlights the open issue raised by the reviewer.

Note the exclusion of nonadiabatic terms in Eq. 10 is consistent with the assumed separability of fast and slow molecular degrees of freedom. While nonadiabatic interactions drive relaxation and are key ingredients in dynamics, we leave for future work to discern their relevance for the equilibrium statistical mechanics of polaritonic materials.

b. Quantum coherent polaritons and macroscopic averaging

In general I think it is necessary to highlight much more that the polaritonic modes of Eq.(11) are assumed quantum coherently delocalized over the full ensemble.

Here, we respectfully disagree with the reviewer. The polariton modes in Eq. (11) are not necessarily quantum coherently delocalized over the full ensemble. This would only be the case in a limit where heterogeneity in the molecular ensemble is negligible. In particular, the polariton frequencies and Hopfield coefficients at each molecule depend, a priori on, $\boldsymbol{\theta}$ and \mathbf{X} of all molecules. Certainly, we expect some modes will be much more strongly localized than others depending on the energy of the mode and disorder (as suggested by our works and those of others). We also want to point out that because the polaritons are being obtained as normal-modes of a Hamiltonian, this does not mean they are delocalized across the entire system (see e.g., Agranovich and Gartstein, PRB, 75, 075302). We continue this comment in our response to the additional points made by the referee below, but in any case, we added the following statement to clarify (page 4):

Note that H_{Pol} , $c_{\mathcal{P}}^{\dagger}$, $c_{\mathcal{P}}$, and $\omega_{\mathcal{P}}$ are all dependent on the molecular orientations $\boldsymbol{\theta}_{\zeta}$, center of mass positions \mathbf{X}_{ζ} and number of molecules of each chemical species contributing to the formation of the polariton modes \mathcal{P} . These eigenmodes may be more or less localized depending on their energy and the typical size of fluctuations of the molecular ensemble (disorder) [refs.].

Taking into account the potential large degeneracies and non-adiabatic behavior of the polaritonic modes I would expect strong decoherence and thus mostly classical behaviour emerging. Indeed, I would expect an analogous behaviour as for the total electronic wave function of the ensemble, which is usually assumed to decohere between the different molecules.

To some extent this seems assumed later when averaging over all rotational and translational degrees of freedom for Q_{pol} above Eq.(16). If I understand correctly the argument, the "mean-field frequencies" and the resulting Q_{pol} are essentially classical/macroscopic quantities. And I would expect that after averaging, the resulting distribution would be much more like a Maxwell-Boltzmann distribution, in contrast to the simplified model in Section III where no averaging procedure seems to be applied.

We respectfully disagree with the reviewer. To support our arguments, we will refer to experimental investigations

of polariton thermal emission spectrum, e.g., ACS Photonics 2018, 5, 1, 217-224 (see erratum) and Nano Letters 2021, 21, 4, 1831-1838. These works (as well as many others including some much earlier ones e.g., this detailed review has plenty of discussion of thermal polaritons: Physics reports 217, no. 4 (1992): 159-223.) provide direct measurements of polariton frequencies (spectral peaks) and (relative) mean occupation numbers (via the spectral intensities) obtained from thermal emission detection.

These references provide strong evidence polaritons satisfy a Bose-Einstein thermal distributions (i.e., not Maxwell-Boltzmann). To be clear, these references provide measurements of polariton thermal emission spectrum in disordered systems which are theoretically reproduced by computations with Kirchoff’s law (in the case of the ACS Photonics article, see the erratum where the authors address specifically the theoretical computations and how ultimately the experiments show agreement with Kirchoff’s law) that essentially imply a Bose-Einstein thermal occupation for polaritons. Note that for modes with high frequencies (relative to room temperature) as molecular vibrational polaritons, the Bose-Einstein distribution at room temperature is dramatically different from the Maxwell-Boltzmann, and therefore, had these experiments considered a “classical” version of Kirchoff’s law, there is no way that the experimental results would match theoretical computations.

Ultimately, these references indicate that polaritons may be short-lived quasiparticles, but in the strong coupling limit, they have a stationary Bose-Einstein thermal distribution.

Our second point addresses the comment by the reviewer on the “mean field frequencies” and the resulting Q_{pol} . To explain our view, we again allude to the experimental observations in the papers cited above. Specifically, we note that while polariton thermal emission spectra are macroscopic quantities following from the averaging of a macroscopic sample with disorder on a microscopic scale (i.e., these experiments are not at all reporting single-photon measurements, and the collection of photons probably spans a macroscopic time window where one can imagine the samples undergo many fluctuations), the theoretical modeling of the input required for application of Kirchoff’s law in those experiments (transfer matrix simulations) included only information about the bare matter phonon modes (in the homogeneous limit for the sample), the photonic material dispersion and an assumed homogeneous (mean-field) light-matter interaction. This shows to us that several properties of polariton modes (their energies and Hopfield coefficients) as represented in Eq. 11 survive the macroscopic averaging unavoidable in thermal measurements.

In light of these comments, in our view, there is no contradiction between Sec. III and the general treatment of Sec. II ultimately leading to Eq. 27 (of this version). Section III is a special case of Sec. II where we retain only a single photon-mode and assume a trivial probability distribution for \mathbf{X} and θ in a 0D geometry where the distances between the molecules are taken as approximately zero and the molecules share the same orientation.

I would highlight that without any averaging procedure the microscopic polaritonic quantum modes do not become macroscopic and hence cannot compete with the otherwise macroscopic energy scaling of the rest of the system. In this case the effect of the polaritons diminishes with increasing system size.

To the extent that we understand the reviewer’s point, we partially disagree that the “microscopic polaritonic quantum modes do not become macroscopic” and refer to our answer to the previous comment above and the experimental references that we cited. As we express in more detail above, simplified Hamiltonians, essentially equivalent to that in Sec. III, can reliably predict (macroscopically obtained) polariton thermal emission spectrum in the collective VSC regime. Our view is the reported decrease of polaritonic effects with increasing system size in Sec. III is due to the large mismatch in density of (hybrid) light-matter states relative to weakly coupled (or dark modes in the idealized Hamiltonian of Sec. III). We expect this issue will be mitigated in a more detailed treatment of molecular and EM degrees of freedom.

Hence I would also not be so sure that the difference between Ref. 3 and the current work (except of in this simplified model) is really due to the Bose-Einstein distribution.

For the reasons described in the comments above, we partially disagree with the reviewer on this point. Our perspective is that at sufficiently high temperatures, we reach the classical limit, and our conclusions agree with those of Li et al who analyzed polaritons using a classical statistical mechanics approach. Certainly, our analysis emphasizes the relevance of other complex features typically neglected in studies of light-matter systems like the multiple EM modes, the strong interaction between potentially multiple bright molecular vibrations and different resonances of the cavity mode, the configurational averaging involved in the macroscopic polaritonic partition function, etc. However, it remains to us the case that a major difference of our work relative to Li et al. is our consideration of quantum

statistics (specifically the Bose-Einstein distribution).

c. Effective few-mode description

Following the above discussion, I think it is important to note that commonly the single- or few-mode approximation is a substitution for taking a part of the continuous photonic density of states into account effectively. Especially if one reduces to the mean-field polaritonic modes, I think it is more the macroscopicity of the molecular ensemble, as also stated in the current manuscript, that is responsible for any observable effect than the fact that a cavity contains a continuum of modes. I would propose to stress this a little more in the current manuscript.

Our view is that the multimodal and macroscopic nature of the electromagnetic field in confined media is as relevant as the macroscopic character of the molecular ensemble. This stems from the fact that the canonical partition function can be written as a Laplace transform of the microcanonical density of states which we expect plays an important role in the thermal description of polaritonic materials. For example, the density of quasiparticle modes of any type tend to have a strong dependence on dimensionality and frequency-momentum dispersion. Further, as we have shown in previous work, even polaritons formed under off-resonance conditions can have markedly different properties relative to pure light and matter states, and therefore, may have a significant effect in thermal properties of polaritonic materials.

These comments are given here for clarification of our view; they do not aim to imply that effective single-mode or few-mode theories are inadequate, as it is likely possible (with e.g., renormalization group theory) to systematically coarse-grain the EM field and potentially obtain an effective theory of the confined EM system modeled by a small number of modes while preserving the effects of the integrated out modes. This is certainly a direction we think is promising, but have not yet seen this. Quite to the contrary, in earlier studies of transport under strong light-matter coupling, we have obtained evidence the (perhaps naive) standard single-mode approach leads to conclusions in qualitative disagreement with a multimode description.

Clarifications:

d. Parametric dependence and polaritonic eigenmodes

I think Eq.(10) is not rewriting the original Hamiltonian but approximating it for the subsequent thermal ensemble considerations. Clearly the Hamiltonian in Eq.(10) is not the same as the one of Eq.(2), which contains also all quantum states for rotations and translations. I also think it would be beneficial if the authors would write specifically in Eq.(11) the parametric dependence on all the coordinates. I think this would be helpful to distinguish the microscopic polaritonic degrees of freedom from the macroscopic ones upon averaging.

We thank the reviewer for pointing these out. The parametric dependence has now been added to Eq. 11. We also added the following statements to clarify that under the assumed separability conditions, the thermodynamic predictions associated to the Hamiltonian in Eq. 2 can be obtained from a statistical mechanical treatment of Eq. 10.

We infer (under the separability conditions and classical correspondence previously delineated), the thermodynamic properties of the total light-matter Hamiltonian (Eq. 2) can be obtained from the statistical mechanical treatment of the effective Hamiltonian (here written in the basis of eigenmodes of $h_M(\mathbf{J}, \mathbf{P}_C) + H_L + H_{LM}(\boldsymbol{\theta}, \mathbf{X})$,

e. Configurational averaging

I am not so sure that the references the authors give for this procedure, i.e. [56,57] in the current manuscript, are the best ones, since they are for electronic strong coupling and based on perturbation theory. Could the authors show or discuss, maybe in an appendix, how such a procedure would be performed. In the end it seems everything relies on a reasonable approximation for the parametrically dependent frequencies and modes. I think it would be good to refer in this context to recent work on this topic 4,5. This would explain better the mean-field frequencies which go into the final expression (24).

We thank the reviewer for recommending us to look at these studies and for the suggestion to further address the configurational averaging. We have added some comments which we expect will clarify our treatment and included citations to works suggested by the reviewers. In particular, we have separated the formal discussion of the macroscopically averaged polaritonic partition function from the discussion of how to approximate this quantity. On page 4, we added (paragraph below Eq. 15):

The polariton partition function $Q_{Pol}(N, V, T)$ is given by the macroscopic average of $Q_{\mathcal{P}}(\boldsymbol{\theta}, \mathbf{X}) = \text{Tr}[\exp(-\beta H_{Pol}(\boldsymbol{\theta}, \mathbf{X}))]$ over the space of molecular positions \mathbf{X} and orientations $\boldsymbol{\theta}$. Assuming the molecular system is isotropic and uniformly distributed (over long distances), it follows that

$$Q_{Pol}(N, V, T) = \frac{1}{h^{6N_M}} \int d\boldsymbol{\theta} d\mathbf{X} Q_{\mathcal{P}}(\boldsymbol{\theta}, \mathbf{X}) \quad (1)$$

$$= \frac{1}{h^{6N_M}} \int d\boldsymbol{\theta} d\mathbf{X} \text{Tr} \exp[-\beta H_{Pol}(\boldsymbol{\theta}, \mathbf{X})]. \quad (2)$$

On page 6, we added the following comments regarding the evaluation of $Q_{Pol}(N, V, T)$ and the generation of mean-field polariton frequencies):

To conclude this discussion of our formalism, we note that, under collective vibrational strong coupling, polariton frequencies $\omega_{\mathcal{P}}$ depend on the orientation and center of mass coordinates of a large number of molecules, and therefore, we expect negligible fluctuations in the spectrum of $H_{Pol}(\boldsymbol{\theta}, \mathbf{X})$ from its macroscopic average. This feature suggests a simple approximation to the light-matter partition function

$$\begin{aligned} Q_{Pol}(N, V, T) &\approx \prod_{\mathcal{P}} \bar{q}_{\mathcal{P}}(T) \\ &= \prod_{\mathcal{P}} \frac{e^{-\beta \hbar \bar{\omega}_{\mathcal{P}}/2}}{1 - e^{-\beta \hbar \bar{\omega}_{\mathcal{P}}}}, \end{aligned} \quad (3)$$

where $\bar{q}_{\mathcal{P}}$ and the corresponding frequencies $\bar{\omega}_{\mathcal{P}}$ are harmonic partition functions and frequencies obtained from the (uniform and isotropic) translational-orientational average of the normal-mode spectrum of the quadratic polariton Hamiltonian (Eq. 11). Several methods can be employed to estimate the mean frequencies $\bar{\omega}_{\mathcal{P}}$ [refs.]. For example, in their study of polariton scattering and localization, Refs. [ref.] and [ref.] obtained macroscopically averaged polariton frequencies in the rotating-wave-approximation and the same methods can be applied to generate mean-field normal-mode frequencies of any positive-definite quadratic Hamiltonian.

f. Definition of $q_F(V, T)$

In Eqs.(22-24) the expression $q_F(V, T)$ appears the first time, I believe. I think it is just the product of the different contributions as defined in Eq(15), but it would be good to unambiguously define all the quantities that appear in (24).

We agree and have added the definition (which is exactly what the reviewer noted) after Eq. (23).

g. Discussion

I think to refer to N_p polaritonic modes and Eq.(11) is not correct, since Q_{pol} is the macroscopically averaged expression. I would again highlight that these averaged modes are macroscopic in nature and hence using a microscopic form will lead to negligible contributions as demonstrated in the example.

We respectfully disagree with the reviewer. The modes in Eq. 11 are obtained from solving the quadratic time-independent light-matter Schrodinger equation for a fixed configuration of the molecular system (center of mass positions and orientations). From our point of view, the experiments on thermal emission described above provide strong evidence that Eq. 11 if employed with a typical random set of molecular orientations (isotropic) and positions (uniformly distributed in microcavity with a fixed density) will lead to polariton eigenmodes described by Eq. 11 which are appropriate for performing estimates of polaritonic effects on thermal partition functions. The rigorous approach indeed consists of performing a proper configurational averaging, but this is already highlighted in the manuscript and we see no contradiction with Eq. 11 which provides a stepping stone towards the macroscopic averaging process. Certainly, Eq. 11 only arises because we our separability and temperature assumptions lead to a quadratic structure to the light-matter interaction and the matter degrees of freedom involved in strong coupling. However, this point is already thoroughly discussed and emphasized in the manuscript, especially in this new version where we incorporated most of the reviewers' suggestions.

A point that would be good to address is the question of resonances. Is the formula (24) capable of capturing the pronounced resonance conditions as observed in experiments. Could the averaging and the parametrical dependence of the polariton modes generate something strongly peaked at a specific frequency? This would boil down to the question of how sensitive the polariton modes are to disorder in the micro-canonical ensemble.

As we have stated in our conclusions, it remains a goal of future work including large-scale simulation to provide a substantive answer to the question of how Eq. 27 (in this version) compares to experiments. Whether and how sharp resonances can occur in polaritonic effects on molecular materials is an intriguing question that we are investigating with the aid of Eq. 27, but have no preliminary answers.

h. Application to single-mode cavity

I think in this and the subsequent discussion it should be highlighted that there seems to be no macroscopic averaging of the polaritonic degrees of freedom. Thus it is unsurprising that for large number of molecules the results diminish. To me it seems that the authors take, instead of many micro-canonical realizations of the mean-field polariton, just one realization. Is this interpretation correct?

The reviewer is correct that in this section we have not performed configurational averaging. We have emphasized this point by adding to the main manuscript the statement below (page 7):

We also ignore disorder effects by assuming a 0D microcavity geometry and a perfectly oriented molecular ensemble. This limit is equivalent to assuming trivial probability distributions (Dirac delta functions) for \mathbf{X}_ζ , θ_ζ and the matter normal-mode frequencies.

We agree that the reduction with size of polariton effects on the probed model is not necessarily surprising, although the scaling is not exactly the $1/N$ that is generally alluded to in the literature, in part because of Eq. 27 which expresses a nontrivial relation between the equilibrium reaction quotients in the presence and absence of strong coupling. This point is stated on page 11 (right before the conclusions).

Minor points:

i. Maybe clarify that the use (k_x, k_y, k_z) is restricted to planar Fabry-Perot cavities, and for general cavity structures there is not necessarily a well-defined momentum associated.

We followed this recommendation by adding to page 2 the statements:

Without loss of generality, we assume the bare field dynamics conserves momentum along the x , y and z directions, so H_L is given by ...

Note other photonic structures could also be treated with Eq. 3 by employing modes defined in terms of suitable quantum numbers according to symmetry and boundary conditions satisfied by the corresponding EM field.

j. Maybe indicate the upper limit in Eq.(7) and make p_α boldfaced below. Is it $N_A + \dots + N_E$? By the way, why E and not D in denoting the species?

We have added the upper limit to Eq. 7 and in the text below explain $N_M = N_A + N_B + N_C + N_E$. We also made \mathbf{p}_α boldfaced in the same paragraph. Here, we want to note the reason we worked with E and not D was that later we employ the subscript D to denote the “dark” part of the Hamiltonian (which describes dynamics that in our approximation is independent of the transverse EM field).

k. In Eq.(6) and below (7) α refers to one particle in the Hamiltonian of species F (maybe indicate this). Later in Eq.(11) alpha refers to the microscopic polariton modes. Maybe choose a different notation.

We have changed the notation so the general polariton label previously given by α (in eqs 6 and 7 and elsewhere) is now denoted by \mathcal{P} .

l. Shouldn't the dependence in j_ζ^{ab} of Θ and \mathbf{Y} be rather Θ_ζ and Y_ζ below Eq.(9)?

We thank the referee for spotting this, and note we have made the suggested correction.

m. Maybe indicate somewhere explicitly that Theta, X, ... are actually multi-dimensional depending on the full ensemble of molecules, i.e., give explicitly the upper limit for zeta. Below Eq.(10) in H_{pol} there should also be a dependence on X. Why is there a zeta dependence on X in $V_M(\Theta, X)$?

We thank the referee for making these points. We have made the following changes to the text to address them: we removed the ζ dependence from Eq. 10. (in particular, V is now written only in terms of \mathbf{X} and $\boldsymbol{\theta}$); we have also made explicit the dependence of H_{Pol} on \mathbf{X} and added statements making it clear the global dependence of $\boldsymbol{\theta}$ and \mathbf{X} on the ensemble (page 3)

...parametrized by the set of center of mass position and momentum of each molecule $(\mathbf{X}_1, \mathbf{P}_{C_1}, \dots, \mathbf{X}_{N_M}, \mathbf{P}_{C_{N_M}}) = (\mathbf{X}, \mathbf{P}_C)$ and their corresponding (classical) orientation and angular momentum $(\boldsymbol{\theta}_1, \mathbf{J}_1, \dots, \boldsymbol{\theta}_{N_M}, \mathbf{J}_{N_M}) = (\boldsymbol{\theta}, \mathbf{J})$

n. Maybe indicate below Eq.(16) that the exponential in the definition of Q_{pol} is macroscopically averaged.

We have reformulated the text right below Eq. 17 as suggested by the reviewer:

where $A_{Pol}(N, V, T) = -k_B T \ln[Q_{Pol}(N, V, T)]$ (with $Q_{Pol}(N, V, T)$ corresponding to the macroscopically averaged polariton contribution to $Q(N, V, T)$ as given by Eq. 16). Likewise applies for $A_D(N, V, T)$, which is the free energy of the modes with dynamics generated by H_D .

o. Above Eq.(18) the "A" of the free energy should not be a subindex. Maybe also briefly comment on how this chemical potential affects the photons, which are usually associated with zero chemical potential.

We thank the referee for pointing out this typo which has now been fixed. Noting that in our description, photons are the same as polaritons with negligible molecular content, we have also added the following comment regarding the polariton contribution to the matter chemical potential:

Note that $\mu_{F, Pol}(N, V, T)$ corresponds to the change in the chemical potential of species F induced by the strong light-matter interaction and is unrelated to the polaritonic chemical potential. This quantity vanishes at thermal equilibrium as follows for any non-conserved quasiparticles [refs.].

We hope our response clarify and address every point raised by the reviewer and we thank them again for their time and attention to our manuscript as well as for the many insightful comments and questions.

Reviewers' Comments:

Reviewer #2:

Remarks to the Author:

The authors have answered my questions satisfactorily and adapted their manuscript accordingly. I think the work can be published in the current form.